# AgentBoard: An Analytical Evaluation Board of Multi-turn LLM Agents

**Chang Ma**$^{*\spadesuit}$    **Junlei Zhang**$^{*\diamondsuit\triangle}$    **Zhihao Zhu**$^{*\heartsuit}$    **Cheng Yang**$^{*\clubsuit}$
**Yujiu Yang**$^{\clubsuit}$    **Yaohui Jin**$^{\heartsuit}$    **Zhenzhong Lan**$^{\triangle}$    **Lingpeng Kong**$^{\spadesuit}$    **Junxian He**$^{\star}$
$^{\spadesuit}$The University of Hong Kong    $^{\diamondsuit}$Zhejiang University    $^{\heartsuit}$Shanghai Jiao Tong University
$^{\clubsuit}$Tsinghua University    $^{\triangle}$ Westlake University    $^{\star}$HKUST
llmagentboard@gmail.com

## Abstract

Evaluating Large Language Models (LLMs) as general-purpose agents is essential for understanding their capabilities and facilitating their integration into practical applications. However, the evaluation process presents substantial challenges. A primary obstacle is the benchmarking of agent performance across diverse scenarios within a unified framework, especially in maintaining partially-observable environments and ensuring multi-round interactions. Moreover, current evaluation frameworks mostly focus on the final success rate, revealing few insights during the process and failing to provide a deep understanding of the model abilities. To address these challenges, we introduce AGENTBOARD, a pioneering comprehensive benchmark and accompanied open-source evaluation framework tailored to analytical evaluation of LLM agents. AGENTBOARD offers a fine-grained progress rate metric that captures incremental advancements as well as a comprehensive evaluation toolkit that features easy assessment of agents for multi-faceted analysis. This not only sheds light on the capabilities and limitations of LLM agents but also propels the interpretability of their performance to the forefront. Ultimately, AGENTBOARD serves as a step towards demystifying agent behaviors and accelerating the development of stronger LLM agents.[1]

## 1   Introduction

General-purpose agents that can autonomously perceive and act in various environments are considered significant milestones in Artificial Intelligence (Russell and Norvig, 2005). Recent advancements in large language models (OpenAI, 2023; Touvron et al., 2023) have demonstrated emergent agent abilities that enable them to understand diverse environments and perform step-by-step planning through multi-round interactions (Yao et al., 2023; Song et al., 2023). These advanced abilities contribute to the potential of LLMs to act as generalist agents for real-world problem-solving.

A comprehensive evaluation of LLM agents is crucial for the progression of this emerging field. To start, *task diversity* is necessary to cover various agent tasks such as embodied, web, and tool agents. Additionally, *multi-round interaction* is critical to mimic realistic scenarios, in contrast to the single-round tasks commonly adopted in existing benchmarks (Xu et al., 2023b; Lin and Chen, 2023; Qin et al., 2023a). Furthermore, evaluating agents in *partially-observable* environments, where they must actively explore to understand their surroundings, is essential for practical assessments. This differs from the "synthetic" agent tasks (Wang et al., 2023b) derived from conventional benchmarks in fully-observable environments, such as MMLU (Lanham et al., 2023) and GSM8K (Cobbe et al., 2021). However, existing agent benchmarks fail to satisfy all of these criteria.

---

$^{*}$Equal Contribution.
[1]Code and data are available at https://github.com/hkust-nlp/AgentBoard

38th Conference on Neural Information Processing Systems (NeurIPS 2024) Track on Datasets and Benchmarks.

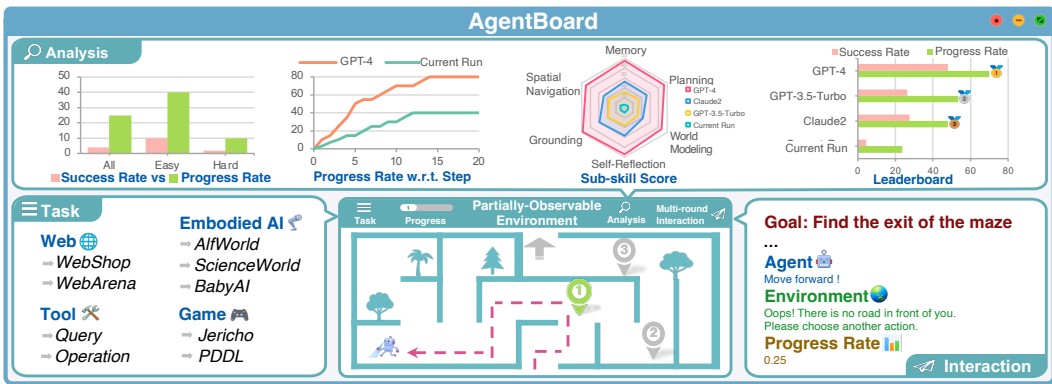

Figure 1: The illustrative overview of AGENTBOARD. AGENTBOARD consists of a 9 diverse tasks. Agents interact in multi-rounds with partially-observable environments to achieve each subgoal. Furthermore, AGENT-BOARD provides an open-source analytical evaluation framework, as shown in the figure.

Moreover, the inherent complexity in agent tasks characterized by multi-round interactions distinguishes them significantly from other language tasks. Due to this complexity, there is a pressing need to delve into the details and gain a deeper understanding of how models function during the process. Nonetheless, most current evaluations predominantly rely on the final success rate as their metric, which provides limited insights into these intricate processes (Liu et al., 2023a; Wang et al., 2023b; Yao et al., 2023; Liu et al., 2023b; Mialon et al., 2023). This simplified evaluation is particularly inadequate in challenging environments where most models demonstrate nearly zero success rates, consequently blurring finer distinctions and obscuring underlying mechanisms (Liu et al., 2023a).

To address these issues, we introduce AGENTBOARD, a benchmark designed for multi-turn LLM agents, complemented by an analytical evaluation board for detailed model assessment beyond final success rates. AGENTBOARD encompasses a diverse set of 9 unique tasks and 1013 exemplary environments, covering a range from embodied AI and game agents to web and tool agents. Each environment, whether newly created or adapted from pre-existing ones, is carefully crafted and authenticated by humans to ensure multi-round and partially observable characteristics in a unified manner. Notably, we have defined or manually annotated subgoals for each data sample, introducing a unified *progress rate* metric to track the agents' detailed advancements. As we will demonstrate in §4.2, this metric uncovers significant progress made by models that would otherwise appear trivial due to negligible differences in success rates.

Along with the benchmark, we develop the AGENTBOARD evaluation framework as an open-source toolkit that features an analytical web panel to examine various dimensions of agent abilities through interactive visualization. The toolkit offers a unified interface, providing users with easy access and effortless customization options. As shown in Figure 1, the AGENTBOARD toolkit currently supports analysis and visualization on fine-grained progress rates tracking, performance breakdown for hard and easy examples, detailed performance across various sub-skills, long-range interaction assessment, grounding accuracy, and trajectory. This detailed evaluation is crucial for acknowledging the progress of LLM agents and for guiding the development of more robust LLM agent models. The comparison between AGENTBOARD and previous works is shown in Table 1.

We evaluated a range of proprietary and open-weight LLM agents using AGENTBOARD, obtaining insights into the current landscape of LLMs as agents. Key findings include: (1) GPT-4, unsurprisingly, outperforms all other models by exhibiting extensive proficiency across distinct agentic abilities with Llama3 (Touvron et al., 2023) and DeepSeek LLM (DeepSeek-AI et al., 2024) taking the lead; (2) Strong LLM agents are characterized by their capability for *multi-turn* interaction with the environment, an ability that is notably lacking in most open-weight models; (3) Emergent agentic abilities are strongly dependent on basic abilities like grounding, world modeling, and self-reflection. Current proprietary models typically demonstrate comprehensive agentic abilities, while open-weight LLMs show varying deficiencies. Through AGENTBOARD, we highlight the importance of analytic evaluation of LLM agents. The detailed evaluations provided by AGENTBOARD and its open-source toolkit are expected to significantly contribute to the further development of LLM agents.

Table 1: AGENTBOARD differs from other LLM benchmarks by providing a comprehensive framework that integrates all four guiding principles within its evaluation system. ‡Notably, AgentBench entails both single and multi-round tasks, with mainly the former differentiating open-sourced models. §The GAIA benchmark focuses solely on question answering tasks. †MINT benchmark primarily includes fully-observable environments tasks derived from conventional evaluations such as HumanEval and GSM8K.

| Benchmarks | Task Diversity | Multi-round Interaction | Partially-Observable Environments | Fine-grained Progress Metrics | Analytical Evaluation |
|---|---|---|---|---|---|
| AgentBench (Liu et al., 2023a) | ✔ | ✗‡ | ✔ | ✗ | ✗ |
| GAIA (Mialon et al., 2023) | ✗§ | ✔ | ✔ | ✗ | ✗ |
| MINT (Wang et al., 2023b) | ✔ | ✔ | ✗† | ✗ | ✗ |
| API-Bank (Li et al., 2023) | ✗ | ✔ | ✔ | ✗ | ✗ |
| ToolEval (Qin et al., 2023b) | ✗ | ✔ | ✔ | ✗ | ✗ |
| LLM-Eval (Lin and Chen, 2023) | ✔ | ✗ | ✗ | ✗ | ✗ |
| AGENTBOARD | ✔ | ✔ | ✔ | ✔ | ✔ |

## 2 AGENTBOARD – Overview

AGENTBOARD is a unified, open-source benchmark for evaluating LLM agents that adheres to five key principles: *task diversity, multi-round interaction, partially-observable environments, fine-grained metrics, and analytical evaluation*, as shown in Table 1. Our commitment to these principles manifests in three key areas:

- *Task Diversity and Uniformity*, where we carefully curate nine diverse environments across four scenarios, ensuring they require multi-round interactions, are fully text-based and primarily partially-observable. This contrasts with many existing LLM agent benchmarks, which are often solvable in a single round, derived from fully-observable tasks like MMLU, or focus on a specific type of task, as demonstrated in Table 1. Compared to AgentBench (Liu et al., 2023a), in particular, our choice of tasks makes AGENTBOARD planning-heavier, where the results on AGENTBOARD well-correlates with the scores on traditional reasoning and coding benchmarks, while the AgentBench scores strongly correlated with scores on the knowledge test MMLU, as illustrated in Ruan et al. (2024). We elaborate on the choice of tasks and their adaptation in §3.

- *Fine-grained Progress Rate*, where AGENTBOARD is the first to propose a fine-grained progress rate metric tracking the intermediate progress of different agents. This metric distinguishes our benchmark in tracking minimal improvement in LLM agent performances. Such a capability is crucial in current endeavors to develop stronger open-weight LLMs, providing detailed insights that are essential for incremental advancements in agent capabilities. We provide detailed introduction for this metric and its annotation process in §2.2.

- *Comprehensive Analysis*, where AGENTBOARD is the first LLM Agent benchmark to expand metrics beyond mere success rate and scores to include detailed analyses. As illustrated in Figure 1, such a comprehensive evaluation includes (1) fine-grained progress rates tracking different agents, (2) grounding accuracy, (3) performance breakdown for hard and easy examples, (4) long-range interactions, (5) analyses of performance across various sub-skills, and (6) trajectory with friendly visualization. We elaborate these analyses in our experiments at §4. Additionally, AGENTBOARD provides an web interface[2] through Wandb dashboard that offers interactive visualizations of these analyses during evaluation. We perform a case study on the panel in §5.

### 2.1 A Unified Multi-Round Reflex Agent

**Preliminaries:** An LLM agent receives textual world descriptions, chooses a text action, and gets feedback detailing state changes and any action errors. Interaction with these environments can be modeled as a special case of Partially Observable Markov Decision Processes (POMDPs) defined by tuple $\langle g, \mathcal{S}, \mathcal{A}, \mathcal{O}, \mathcal{T} \rangle$, with goal $g$, state space $\mathcal{S}$, valid actions space $\mathcal{A}$, observation space (including environment feedback) $\mathcal{O}$, transition function $\mathcal{T} : \mathcal{S} \times \mathcal{A} \to \mathcal{S}$. An agent with policy $\pi$ makes prediction at time step $t$ based on goal $g$ and memory $m_t = \{o_j, a_j, o_{j+1}, a_{j+1}, \ldots o_t\}, 0 \le j < t$, which is a sequence of actions and observations. This trajectory of the agent $\tau = [s_0, a_0, s_1, a_1, \ldots s_t]$ is formulated by policy and environmental state transitions, such as

$$p_\pi(\tau) = p(s_0) \prod_{t=0}^{T} \pi(a_t|g, s_t, m_t) \mathcal{T}(s_{t+1}|s_t, a_t) \tag{1}$$

---

[2]An example of the panel is public here.

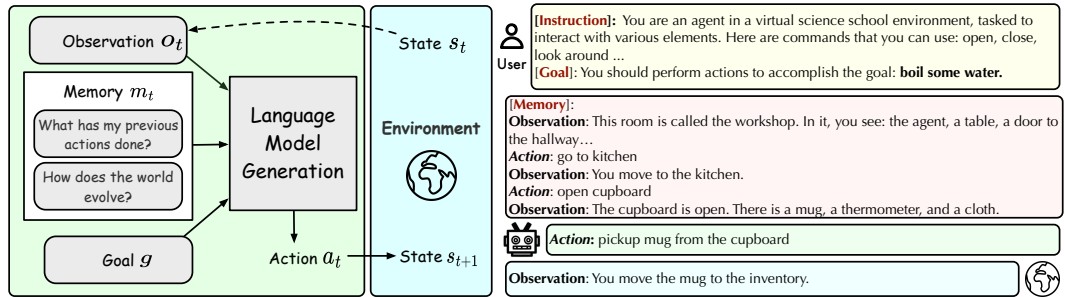

Figure 2: (Left) A structural overview of the reflex agent, which iteratively interacts with the environment and makes next step predictions based on the goal and history. (Right) An example of a prompt for our reflex agent.

Table 2: Examples of goals for the 4 task categories in AGENTBOARD, along with a sampled step of the trajectory and progress rate. The trajectory is generated by GPT-4. Some lengthy observations are omitted with "..." for brevity. The task name in the table uses an abbreviation, the full name can be found in §3.

| Task | Goal & Trajectory |
|------|-------------------|
| ALF | **Goal: put a clean egg in microwave.**
Step 02    **Action:** open fridge 1   **Observation:** You open the fridge 1. The fridge 1 is open. In it, you see a apple 2, a egg 1, a lettuce 1, a pan 2, a plate 1, and a tomato 1.    Progress Rate: 0.00 → 0.25 |
| JC | **Goal: Get out of the house. Then escape the city without getting caught via driving.**
Step 29    **Action:** take a shower   **Observation:** You step into the shower, turn on the water, and within a few moments you feel like a new man. But no time to dawdle - you hop back out again and dry off in record time . . .    Progress Rate: 0.43 → 0.57 |
| WA | **Goal: Display the list of issues in the kkroening/ffmpeg-python repository that have labels related to questions**
Step 05    **Action:** click [5398]   **Observation:** Tab 0 (current): Issues · Karl Kroening / ffmpeg-python · GitLab [6573] RootWebArea 'Issues · Karl Kroening / ffmpeg-python · GitLab' focused: True [6620] link . . .    Progress Rate: 0.25 → 0.50 |
| TO | **Goal: In "Sheet17", calculate and complete the "Profit" of the products in the table based on the sales information of the products. And then, sort the table in descending order by "Profit".**
Step 07    **Action:** update_cell_by_formula with Action Input: {"operator": "PRODUCT", "start_position": "C8", "end_position": "D8", "result_position": "E8"}   **Observation:** [['Product', 'Category', . . .    Progress Rate: 0.47 → 0.49 |

**The Unified Framework:** AGENTBOARD unifies all tasks around a general framework where the agent receives observations $o_t$ and performs actions $a_t$, causing deterministic state transitions $\mathcal{T} : (s_t, a_t) \rightarrow s_{t+1}$ based on real-world dynamics. A feedback function $f$ is also defined in the environment to derive feedback from each interaction round $o_t = f(s_t, a_t)$. This feedback includes: (1) list all valid actions when the agent uses help actions such as *check valid actions*; (2) execute valid action $a_t$ and return a description of the changed stete $s_{t+1}$; (3) issue errors when the agent performs an action outside of the action space.

We aim to use a simplistic agent framework to showcase LLM basic agentic abilities. As shown in Figure 2, our agent makes decisions based on its memory of past perceptions, similar to how humans learn from experience and adapt. The implementation of the reflex agent assessed in this paper adopts an act-only prompting strategy in line with recent studies (Liu et al., 2023b; Zhou et al., 2023; Xu et al., 2023b), detailed in the right part of Figure 2, while other prompting strategies can be easily incorporated into our open-source framework. Also, LLM agents tend to struggle with limited context lengths in long interactions, failing to retain full history. Following the *"sliding window"* method from LangChain (Chase, 2022), we focus on recent, more impactful interactions (Puterman, 1990) within context constraints. This differs from previous practices that stop the agent when context limits are surpassed (Liu et al., 2023a; Wang et al., 2023b), allowing for extended, intricate interactions in our approach. We provide ablation results such as ReAct prompting (Yao et al., 2023) and other long-context processing techniques to justify our framework in Appendix F.

## 2.2 Fine-grained Progress Rate

Recent studies highlight the predominant use of success rate as the main metric for agent evaluation, which fails to capture the nuances of partial task completion by language model agents (Liu et al., 2023a; Li et al., 2023). This approach does not differentiate between near-complete tasks and minimal task execution, treating both as equivalent failures. Alternative metrics like reward scores are available but lack standardization (Chevalier-Boisvert et al., 2019; Wang et al., 2022). To mitigate this issue, we introduce a *progress rate* metric to accurately reflect LM agents' goal attainment at various stages.

In each round of interaction, a progress rate, denoted as $r_t$, is assigned to evaluate the agent's advancement towards the goal state $g$. As the agent moves through the states $\mathbf{s_t} = [s_0, \ldots, s_t]$, we assess its progress using a matching score $f(\cdot, g) \to [0, 1]$ that quantifies the similarity between the current state and the goal state. The initial value of $r_t$ is set to 0, indicating no progress. The progress rate $r_t$ reflects the highest matching score achieved, reaching 1 when the task is completed. The progress rate is formulated as below:

$$
r_t = \begin{cases} r_t^{\text{match}} = \max_{i, 0 \le i \le t} f(s_i, g), & \text{if } f(\cdot, g) \text{ is continuous} \\ r_t^{\text{subgoal}} = \max_{i, 0 \le i \le t} \left( \frac{1}{K} \sum_{k=1}^{K} f(s_i, g_k) \right), & \text{otherwise} \end{cases} \tag{2}
$$

The function $f(\cdot, g)$ measures state similarity in tasks, such as comparing table states in manipulation activities. It works well for tasks with direct state comparisons but is less effective for tasks with ambiguous intermediate states, where progress is hard to measure. We mitigate this by introducing a discrete matching score to assess how closely intermediate states align with defined subgoals. We begin by decomposing the overall goal $g$ into a sequence of subgoals $\mathbf{g} = [g_1, \ldots, g_K]$, with each subgoal leading into the next. The authors manually label each subgoal, which is then checked and adjusted through a rigorous process described in §3.2. Notably, we manually edit the problems for a simpler setup where each final goal aligns with a unique subgoal sequence, and this affects only 5% of the original problems (a detailed descriptions for our adaptations are in Appendix L.1). Note that while we maintain a unique subgoal sequence, this allows for a diverse set of trajectories, e.g. taking detours when accomplishing the task. As an example, for task "clean an egg and put it in microwave", the necessary subgoals would be "open the fridge" → "taking an egg from the fridge" → "clean the egg with sinkbasin" → "put the egg in the microwave". Each subgoal $g_i$ is associated with a labeled state that indicates its completion. To evaluate the match between an agent state and a subgoal, we employ a regular-expression-based matching function denoted as $f(\cdot, g_i) \to \{0, 1\}$ and the progress rate as $r_t^{\text{subgoal}}$ in Equation 2.

We employ progress rate along with the commonly used success rate metric, which computes the proportion of tasks completed within $T$ interactions.

## 3  AGENTBOARD – Task Composition

AGENTBOARD features four task scenarios: embodied, game, web, and tool. These tasks are selected for their diversity and relevance to everyday activities, offering broader scenario coverage than tool-using benchmarks like MINT (Wang et al., 2023b) and ToolEval (Qin et al., 2023b). We specifically select tasks that require multi-round interactions in partially-observable environments, creating a realistic and challenging setting for agents. Examples of goals and trajectories are displayed in Table 2, and task statistics are summarized in Table 14. Further details on the environments and annotation are provided in Appendix K and L.

### 3.1  Environments

***Embodied* - AlfWorld (ALF) (Shridhar et al., 2021)** are household tasks that require agents to explore surroundings and perform commonsense tasks like "put two soapbars in garbagecan". This task uses subgoal-based progress rates (Appendix L.2) and the original success rate of the environment as metrics.

***Embodied* - ScienceWorld (SW) (Wang et al., 2022)** is a challenging interactive text environment testing scientific commonsense, e.g. "measure the melting point of the orange juice". The current subgoals provided by SW do not accurately reflect a language model's performance due to their sparsity and uneven weighting, as further explained in Appendix L.3. To rectify this, we re-annotate the subgoals for calculating progress rate $r_t^{\text{subgoal}}$. We also re-annotate instructions on tool usage and rooms to explore to ensure the uniqueness of subgoal sequence for task completion.

***Embodied* - BabyAI (BA) (Chevalier-Boisvert et al., 2019)** is an interactive 20x20 grid environment where agents navigate and interact with objects within a limited sight range. The original setup uses image-based observations and tensor-based actions like "0: move left". We adapted it to include a textual action space and descriptive textual observations. Furthermore, we re-annotate subgoals for progress rates to fix subgoal sparsity problem in the original environment (Appendix L.4).

***Game* - Jericho (JC) (Hausknecht et al., 2020)** is a collection of text-based game environments staged in fictional worlds. This task is unique as it requires strong world modeling ability : agents could only gain information about the magic world through exploration and interaction. The original games are too long (need 50-300 steps to finish for LLM agents with fixed context length. Therefore we rewrite the goal of each adventure to restrict the games to be finished within 15 subgoals.

***Game* - PDDL (PL) (Vallati et al., 2015)** is a set of strategic games defined with Planning Domain Definition Language (PDDL). We selected 4 representative games, *Gripper, Barman, Blocksworld, Tyreworld* to benchmark

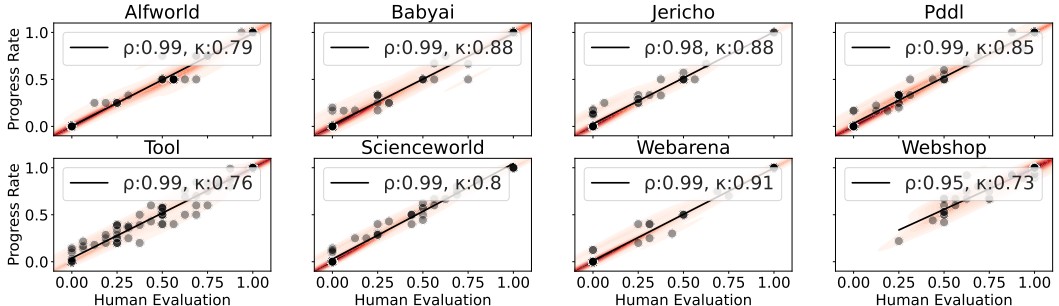

Figure 3: Human verification of AGENTBOARD progress rate. The Pearson correlation coefficients $\rho$ compare human evaluations with the progress rates on 60 different trajectories per task. The Fleiss' kappa $\kappa$ reflects inter-annotator agreement. The trajectories are generated by three models: GPT-4, GPT-3.5-Turbo, and DeepSeek-67b.

LLM agents in diverse scenarios. We adapted the environment implementation (Silver and Chitnis, 2020) written in PDDL expressions to provide text-based observations for agents, enabling a natural language interface for LLM. We measure progress using a matching score, $r_t^{\text{match}}$, which assesses similarity between the current and goal states (Appendix L.6).

*Web* - **WebShop (WS)** **(Yao et al., 2022)** is a network-based simulation environment for e-commerce experiences. Based on the original implementation method (Yao et al., 2022; Shinn et al., 2023), we have improved the error feedback, including refining the observation for exceeding page limits and interacting with wrong objects. These enhancements contribute to the effective interaction of the LLM agent with the environment. We also measure the distance of the current state to the final goal as the progress rate and expand the product scoring rules from Yao et al. (2022) to derive the score (Appendix L.7).

*Web* - **WebArena (WA)** **(Zhou et al., 2023)** is a real web environment featuring various scenarios including business content and discussion forums. To obtain the progress rate, we revised the existing method for calculating the final score (Zhou et al., 2023) and continuously computed the progress rate at each step, fusing the URL matching score with the content matching score, as detailed in Appendix L.8.

*Tool* - **Tool-Query (TQ)** consists of three sub-environments: Weather, Movie and Academia Environment. This tasks primarily involves querying information from respective databases by planning the use of diverse query tools. We manually curate diverse problems for each environment. We also annotate subgoals to compute the progress rate $r_t^{\text{subgoal}}$ (Appendix L.9). While LLMs often provide direct answers in question answering, we only consider an answer correct if the model follows the appropriate trajectory to access the databases. To ensure this, we design questions that cannot be answered directly by state-of-the-art LLMs and provide in-context examples to guide the LLM in querying the databases effectively.

*Tool* - **Tool-Operation (TO)** includes two sub-environments: Todo list management and Google Sheet Operations. These tasks involve using tools to access and modify information. The progress rate in the Todo Environment is measured using $r_t^{\text{subgoal}}$, similar to Tool-Query Environments. In the Sheet Environment, progress is evaluated using $r_t^{\text{match}}$, which uses a matching score between the cells of the current and golden table (Appendix L.10).

## 3.2 Annotation Verification and Metric Justification

After human annotation, we manually verified our labeled subgoals through multiple verification stages to ensure its quality, as detailed in Appendix J. More importantly, we conduct a user study to justify our proposed progress rate metric, asking human annotators to assess the progress of model trajectories and then evaluating its correlation with our automatic progress rate metric. Specifically, we gather 60 model trajectories for each of the 8 tasks from three strong LLMs——GPT-4, GPT-3.5-Turbo, and Deepseek-67b. Each trajectory is assessed by four authors of the paper. The individual human rater is asked to select progress score from $\{0\%, 25\%, 50\%, 75\%, 100\%\}$ given trajectories and task descriptions, without seeing the automatic score. Mean of four scores is taken as the final human score for every trajectory. We show the Pearson correlation between human progress score and the progress rate in Figure 3, and report Fleiss'kappa $\kappa$ to reflect inter-annotator agreement. Results show that progress rate highly correlates with human assessment on the progress where the Pearson correlation exceeds 0.95 on all tasks, and substantial agreement is reached among the annotators.

# 4 Experiments

We conduct a comprehensive evaluation of popular LLMs, including proprietary and open-weight models. Firstly, we report the success rate and progress rate of these agents. Then, we perform detailed analysis of the

Table 3: Performance of different LLMs sorted by success rate. The "Avg." represents the average calculated over all tasks in 9 environments. "A/B" indicates that both "progress rate" and "success rate" are reported. Colors denote proprietary models , open-weight general LLMs , and agent LLMs . Confidence Intervals are reported in Appendix D due to space constraint. ‡ Note that Gemini1.5-Flash's performance may be lower due to stringent security content screening.

| Model | Embodied AI | | | Game | | Web | | Tool | | Avg. |
| | ALF | SW | BA | JC | PL | WS | WA | TQ | TO | |
|---|---|---|---|---|---|---|---|---|---|---|
| GPT-4 | **65.5/43.3** | **78.8/52.2** | **70.7/56.2** | **52.4/35.0** | **81.2/61.7** | **76.5/39.0** | **39.4/15.1** | **85.1/68.3** | **80.8/60.0** | **70.0/47.9** |
| Claude2 | 34.1/24.6 | 32.0/11.1 | 48.1/37.5 | 20.4/ 0.0 | 61.4/40.0 | 74.6/37.8 | 36.4/ 8.6 | 73.5/48.3 | 59.6/27.5 | 48.9/26.2 |
| Gemini1.5-Flash ‡ | 40.9/15.7 | 17.8/ 4.4 | 50.6/38.4 | 11.1/ 0.0 | 23.5/ 6.7 | 72.3/24.7 | 32.4/11.1 | 73.9/46.7 | 68.6/37.5 | 43.5/20.6 |
| Claude3-Haiku | 20.7/ 2.2 | 43.7/14.4 | 34.4/24.1 | 34.7/10.0 | 31.9/13.3 | 73.6/30.3 | 26.8/12.7 | 61.9/24.0 | 62.4/27.5 | 43.3/17.6 |
| Llama3-70b | 29.6/12.7 | 30.4/ 7.8 | 41.1/27.7 | 16.0/ 5.0 | 32.2/20.0 | 74.6/29.9 | 35.6/12.6 | 52.6/36.7 | 65.2/30.0 | 41.9/20.2 |
| GPT-3.5-Turbo | 35.6/17.2 | 31.9/18.9 | 51.7/39.3 | 19.9/ 5.0 | 25.0/ 5.0 | 76.4/35.1 | 25.5/ 4.6 | 69.4/45.0 | 37.2/ 7.5 | 41.4/19.7 |
| xLAM-70b | 53.4/42.5 | 15.4/ 1.1 | 37.7/28.6 | 16.2/ 5.0 | 38.4/16.7 | 73.6/32.7 | 34.5/11.3 | 66.5/38.3 | 26.5/7.5 | 40.2/20.4 |
| DeepSeek-67b | 34.5/20.9 | 36.1/10.0 | 31.7/22.3 | 13.7/ 0.0 | 22.0/ 6.7 | 72.7/31.9 | 23.9/ 5.7 | 71.4/40.0 | 40.5/17.5 | 38.5/17.2 |
| Text-Davinci-003 | 18.8/ 9.0 | 28.9/ 7.8 | 17.5/14.3 | 28.6/10.0 | 31.7/11.7 | 72.3/29.5 | 16.2/ 2.5 | 65.0/38.3 | 56.2/22.5 | 37.2/16.2 |
| GPT-3.5-Turbo-16k | 25.2/ 4.5 | 2.2/ 0.0 | 45.1/33.9 | 16.1/ 0.0 | 22.6/ 3.3 | 73.8/27.9 | 23.7/ 6.1 | 59.1/31.7 | 39.6/15.0 | 34.2/13.6 |
| AgentLM-70b | 58.4/50.7 | 13.0/ 1.1 | 38.0/27.7 | 8.8/ 0.0 | 13.0/ 3.3 | 72.9/31.1 | 13.0/ 5.3 | 50.5/13.3 | 31.8/ 0.0 | 33.3/14.7 |
| Lemur-70b | 10.8/ 0.7 | 33.4/ 5.6 | 19.4/ 9.8 | 10.1/ 0.0 | 9.7/ 3.3 | 71.8/11.6 | 12.2/ 3.3 | 72.0/28.3 | 37.7/12.5 | 30.8/ 8.3 |
| CodeLlama-34b | 11.3/ 3.0 | 3.5/ 0.0 | 19.9/13.4 | 15.5/ 0.0 | 18.5/ 3.3 | 71.7/23.5 | 21.2/ 4.1 | 60.0/13.3 | 48.8/ 7.5 | 30.0/ 7.6 |
| Llama3-8b | 14.1/ 0.7 | 38.8/10.0 | 36.7/22.3 | 10.4/ 0.0 | 20.1/ 3.3 | 68.7/17.5 | 8.3/ 1.6 | 44.2/ 0.0 | 28.7/ 0.0 | 30.0/ 6.2 |
| CodeLlama-13b | 13.4/ 2.2 | 9.6/ 2.2 | 22.2/17.0 | 0.0/ 0.0 | 9.3/ 1.7 | 65.5/25.9 | 17.7/ 3.7 | 52.5/25.0 | 41.8/12.5 | 25.8/10.0 |
| Llama2-70b | 13.2/ 3.0 | 2.6/ 0.0 | 30.0/19.6 | 7.8/ 0.0 | 8.1/ 1.7 | 53.6/13.1 | 11.6/ 3.3 | 48.3/ 0.0 | 38.6/ 0.0 | 23.8/ 4.5 |
| Mistral-7b | 9.8/ 0.0 | 15.8/ 2.2 | 20.1/14.3 | 11.0/ 0.0 | 4.7/ 0.0 | 68.2/13.9 | 13.2/ 1.3 | 51.0/ 3.3 | 27.2/ 0.0 | 24.6/ 3.9 |
| Vicuna-13b-16k | 11.0/ 1.5 | 14.1/ 2.2 | 14.3/ 5.4 | 15.2/ 0.0 | 7.2/ 1.7 | 73.3/21.9 | 11.3/ 2.9 | 34.3/ 3.3 | 26.9/ 0.0 | 23.1/ 4.3 |
| Llama2-13b | 7.8/ 0.0 | 1.1/ 0.0 | 18.1/ 6.2 | 3.2/ 0.0 | 4.1/ 0.0 | 63.5/10.8 | 7.9/ 2.0 | 35.1/ 0.0 | 29.3/ 0.0 | 18.9/ 2.1 |

performance of agents and measure the various abilities of LLM agents, as part of the AGENTBOARD evaluation automatically supported by our open-source toolkit.

## 4.1 Evaluation Setup

We implement the agent as described in §2.1. We use a one-shot in-context example in our prompt, in addition to task instructions. For the detailed prompt, please refer to Appendix N. We benchmark a series of strong proprietary and open-weight models. For open-weight models, we assess the corresponding chat version of them. Please refer to Appendix I for detailed setup.

## 4.2 Main Results

**Progress Rate is more informative and discriminative than success rate.** The success rate and progress rate across various tasks and categories are presented in Table 3. Regarding the overall performance, the progress rate serves as a more effective differentiator between models. For example, Llama2-13b and Mistral-7b exhibit similarly negligible success rates (2.1% and 3.9%, respectively), but their progress rates differ significantly: 18.9% for Llama2-13b and 24.6% for Mistral-7b. This disparity suggests that Mistral-7b generally outperforms Llama2-13b. For models with substantial differences in success rates, such as Text-Davinci-003 outperforming Llama2-70b by 11.7% in success rate, Text-Davinci-003 leads the progress rate by 13.4% as well, which indicates the consistency in performance disparity between significantly different models. Investigating the agent performance on specific tasks, progress rate is often able to differentiate models that have similar success rates – for instance, on the Embodied AI and Game categories, the success rates of most of the open-weight models are similarly low, while they are able to make meaningfully different progresses. Also, the success rate can be influenced by specific characteristics of agents, for example, an agent like CodeLlama-34b often fails to generate the action "finish" when performing tool-using tasks, leading to a higher progress rate and lower success rate compared to CodeLlama-13b. In contrast, progress rate is less susceptible to these agent-specific features as it reflects the overall ability of the agent at each step.

**Proprietary models outperform the open-weight ones.** The performances of *general* LLMs as agents overall follow the scaling law (Kaplan et al., 2020). Larger LLMs outperform their smaller counterparts in most tasks. For instance, the 70 billion parameter models, such as Llama3-70b, DeepSeek-67b, and Lemur-67b, demonstrate superior performance compared to the 7-13 billion parameter models like Mistral-7b and CodeLlama-13b. Notably proprietary models still outperform the best open-weight models: GPT-4 significantly surpasses other LLMs, achieving an average progress rate of 70.0%, followed by Claude and Gemini.

**Strong coding skills help agent tasks.** In the realm of open-weight LLMs, Code LLMs demonstrate a notable advantage compared to other open-weight models: For instance, CodeLlama-34b outperforms Llama2-70b by 6.2% in terms of progress rate, while the significantly smaller CodeLlama-13b surpasses Llama2-70b by 2%. Lemur-70b, which is continual pretrained on code, also significantly surpasses Llama2-70b. This suggests that incorporating a greater volume of code in training data may enhance performance in agent tasks. Additionally,

Table 4: Grounding accuracy (%) on different categories of tasks.

| Model | Embodied AI | | | Game | | Web | | Tool | | Avg. |
|---|---|---|---|---|---|---|---|---|---|---|
| | ALF | SW | BA | JC | PL | WS | WA | TQ | TO | |
| GPT-4 | **82.6** | 22.8 | **80.3** | **100.0** | **93.0** | **98.3** | **97.6** | **97.5** | **98.5** | **85.6** |
| Claude2 | 57.4 | 11.2 | 61.6 | 98.2 | 71.2 | 95.9 | 83.9 | 93.7 | 92.3 | 73.9 |
| Gemini1.5-Flash | 55.9 | 4.3 | 62.9 | 61.3 | 30.3 | 94.5 | 92.7 | 96.6 | 90.8 | 65.5 |
| Claude3-Haiku | 42.6 | 15.5 | 41.2 | 98.2 | 35.4 | 98.2 | 60.0 | 98.2 | 94.7 | 64.9 |
| Llama3-70b | 39.3 | 21.1 | 44.6 | 99.8 | 40.1 | 90.8 | 92.6 | 74.3 | 88.3 | 65.7 |
| GPT-3.5-Turbo | 59.2 | 18.7 | 62.4 | 99.8 | 66.0 | 90.2 | 91.3 | 97.7 | 91.8 | 75.2 |
| Deepseek-67b | 43.6 | 12.6 | 65.4 | 99.8 | 62.7 | 95.4 | 55.7 | 93.1 | 80.5 | 67.6 |
| Text-Davinci-003 | 27.3 | 17.2 | 15.9 | 97.9 | 72.6 | 97.4 | 23.7 | 95.2 | 82.5 | 58.9 |
| GPT-3.5-Turbo-16k | 57.4 | 9.2 | 73.3 | **100.0** | 77.6 | 96.6 | 81.3 | 98.0 | 92.2 | 76.2 |
| Lemur-70b | 15.7 | **47.8** | 44.6 | 97.7 | 31.0 | 84.0 | 82.8 | 96.5 | 88.4 | 65.4 |
| CodeLlama-34b | 8.4 | 0.4 | 28.0 | 97.3 | 43.6 | 98.3 | 96.8 | **97.5** | 82.8 | 61.5 |
| Llama3-8b | 18.1 | 20.4 | 61.4 | 98.2 | 46.2 | 93.1 | 5.1 | 98.4 | 94.7 | 59.5 |
| CodeLlama-13b | 15.8 | 9.0 | 34.6 | 99.7 | 17.8 | 78.0 | 82.1 | 90.5 | 79.2 | 56.3 |
| Llama2-70b | 20.8 | 3.0 | 42.3 | 96.7 | 30.6 | 59.3 | 93.6 | 90.6 | 70.4 | 56.4 |
| Mistral-7b | 12.1 | 7.9 | 33.9 | 96.2 | 18.1 | 83.5 | 37.5 | 69.7 | 35.1 | 43.8 |
| Vicuna-13b-16k | 17.2 | 24.1 | 74.5 | **100.0** | 59.2 | 94.1 | 58.7 | 97.9 | 92.2 | 68.7 |
| Llama2-13b | 8.9 | 8.6 | 37.4 | 99.5 | 56.7 | 73.5 | 79.2 | 86.9 | 74.1 | 58.3 |

Table 5: Progress Rate and Success Rate for easy and hard cases. All models show distinct drop for hard cases.

| Model | Metric | Embodied AI | | Game | | Web | | Tool | | Avg. | |
|---|---|---|---|---|---|---|---|---|---|---|---|
| | | Easy | Hard | Easy | Hard | Easy | Hard | Easy | Hard | Easy | Hard |
| GPT-4 | Progress | 90.6 | 57.4 $\downarrow_{33.2}$ | 70.3 | 62.6 $\downarrow_{7.7}$ | 60.8 | 55.1 $\downarrow_{5.7}$ | 89.3 | 78.5 $\downarrow_{10.8}$ | 79.2 | 62.7 $\downarrow_{16.5}$ |
| | Success | 85.0 | 24.9 $\downarrow_{60.1}$ | 54.2 | 43.3 $\downarrow_{10.9}$ | 32.2 | 21.8 $\downarrow_{10.4}$ | 81.1 | 52.1 $\downarrow_{29.0}$ | 65.6 | 34.4 $\downarrow_{31.2}$ |
| GPT-3.5-Turbo | Progress | 48.8 | 31.0 $\downarrow_{17.8}$ | 31.4 | 10.5 $\downarrow_{20.9}$ | 49.7 | 50.4 $\uparrow_{0.7}$ | 58.3 | 48.8 $\downarrow_{9.5}$ | 34.7 | 34.7 $\downarrow_{12.5}$ |
| | Success | 39.9 | 11.2 $\downarrow_{28.7}$ | 9.2 | 0.0 $\downarrow_{9.2}$ | 25.0 | 14.4 $\downarrow_{10.6}$ | 40.6 | 14.5 $\downarrow_{26.1}$ | 29.9 | 10.1 $\downarrow_{19.8}$ |
| Llama3-70b | Progress | 40.5 | 23.8 $\downarrow_{16.7}$ | 30.3 | 17.0 $\downarrow_{13.3}$ | 53.6 | 55.0 $\uparrow_{1.4}$ | 67.6 | 52.1 $\downarrow_{15.5}$ | 46.6 | 34.4 $\downarrow_{12.2}$ |
| | Success | 29.4 | 3.6 $\downarrow_{25.8}$ | 16.1 | 8.3 $\downarrow_{7.8}$ | 23.3 | 18.1 $\downarrow_{5.2}$ | 48.6 | 21.4 $\downarrow_{27.2}$ | 29.5 | 11.5 $\downarrow_{18.0}$ |
| DeepSeek-67b | Progress | 35.8 | 29.1 $\downarrow_{6.7}$ | 28.8 | 3.8 $\downarrow_{25.0}$ | 50.3 | 45.5 $\downarrow_{4.8}$ | 61.0 | 52.1 $\downarrow_{8.9}$ | 43.1 | 32.2 $\downarrow_{10.9}$ |
| | Success | 26.5 | 7.9 $\downarrow_{18.6}$ | 5.6 | 2.1 $\downarrow_{3.5}$ | 22.0 | 16.6 $\downarrow_{5.4}$ | 40.1 | 20.1 $\downarrow_{20.0}$ | 23.9 | 11.3 $\downarrow_{12.6}$ |
| Lemur-70b | Progress | 26.0 | 15.0 $\downarrow_{11.0}$ | 16.0 | 2.1 $\downarrow_{13.9}$ | 46.1 | 39.1 $\downarrow_{7.0}$ | 59.5 | 51.1 $\downarrow_{8.4}$ | 35.7 | 25.5 $\downarrow_{10.2}$ |
| | Success | 9.2 | 0.3 $\downarrow_{8.9}$ | 2.8 | 0.0 $\downarrow_{2.8}$ | 10.7 | 7.1 $\downarrow_{3.6}$ | 31.4 | 11.8 $\downarrow_{19.6}$ | 13.0 | 4.3 $\downarrow_{8.7}$ |
| CodeLlama-34b | Progress | 13.8 | 6.6 $\downarrow_{7.2}$ | 28.0 | 3.5 $\downarrow_{24.5}$ | 48.3 | 44.5 $\downarrow_{3.8}$ | 66.2 | 45.7 $\downarrow_{20.5}$ | 36.3 | 23.0 $\downarrow_{13.3}$ |
| | Success | 7.2 | 0.9 $\downarrow_{6.3}$ | 2.8 | 0.0 $\downarrow_{2.8}$ | 19.6 | 8.7 $\downarrow_{10.9}$ | 19.2 | 3.6 $\downarrow_{15.6}$ | 11.6 | 3.0 $\downarrow_{8.6}$ |
| Llama2-70b | Progress | 13.6 | 11.0 $\downarrow_{2.6}$ | 13.4 | 1.2 $\downarrow_{12.2}$ | 38.4 | 27.4 $\downarrow_{11.0}$ | 45.9 | 41.8 $\downarrow_{4.1}$ | 26.2 | 19.3 $\downarrow_{6.9}$ |
| | Success | 8.5 | 1.2 $\downarrow_{7.3}$ | 1.4 | 0.0 $\downarrow_{1.4}$ | 12.4 | 3.9 $\downarrow_{8.5}$ | 0.0 | 0.0 $\rightarrow_{0.0}$ | 5.9 | 1.3 $\downarrow_{4.6}$ |

training on code not only benefits tasks that involve using tools, where writing function calls is required, but also improves performance in the Games category, which demands robust planning abilities. This indicates that training on code data could enhance general agentic capabilities beyond code generation.

**Learning on agent tasks further improves performances.** *Agent* LLMs show strong performance among open-weight models. Both `AgentLM-70b` and `xLAM-70b` are specifically trained on agent instruction tuning data. `xLAM-70b` is one of the strongest open-weight models with a success rate of 20.4% surpassing `GPT-3.5-Turbo`. `AgentLM-70b`, trained on `Llama2-70b`, improves by 9.5% in terms of progress rate and 10.2% in terms of success rate. Despite `AgentLM-70b` was trained on trajectories from AlfWorld and WebShop, it also demonstrated significant improvement on other tasks.

## 4.3 Analytical Evaluation in AGENTBOARD

AGENTBOARD provides various analytical evaluations for in-depth understanding of agents as a toolkit. In this section, we'll use this framework to analyze benchmarked models, with all analyses supported by our toolkit via interactive visualizations on the wandb web panel.

**Grounding accuracy.** Grounding is the process of mapping high-level plans to *executable* actions. Errors in grounding valid actions highlight a limitation in the model's ability to follow instructions and format actions correctly, as noted by Zheng et al. (2024). Table 4 reports grounding accuracy, the percentage of valid actions. While `Text-Davinci-003` and `DeepSeek-67b` have lower grounding accuracy than `GPT-3.5-Turbo-16K`, they perform better in main results, indicating strengths in other areas. Specifically, `Text-Davinci-003` shows a grounding accuracy of only 58.9% on average but performs comparably to `GPT-3.5-Turbo` in main results. This suggests that while the model struggles with tool utilization, it excels in planning and other sub-skills. Open-weight models generally have lower grounding accuracy than proprietary ones. Interestingly, `Vicuna-13b-16k`, despite lower main results, achieves a grounding score of 68.7%, comparable to `DeepSeek-67b` and `Claude2`. This underlines why instruction tuning alone couldn't enhance agentic abilities, as found in previous work (Wang et al., 2023b). While tuning improves models' ability to follow instructions, it doesn't necessarily boost overall performance.

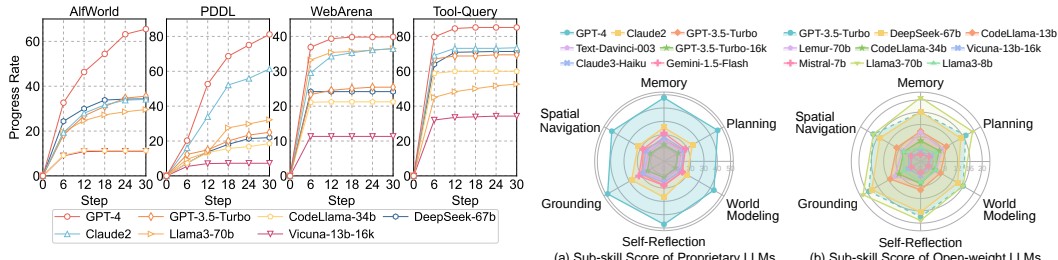

Figure 4: Long-range interaction analysis. Specifically, we report the progress rate w.r.t. step of AlfWorld, PDDL, WebArena and Tool-Query.

Figure 5: The Sub-skill scores of different LLMs.

**Performance breakdown for hard and easy examples.** For each task, we divide environments into "hard" or "easy" based on the number of subgoals/conditions to meet, as shown in Table 14. The outcomes are presented in Table 5. Unsurprisingly, all models show a significant performance drop on hard examples, consistent with Dziri et al. (2023)'s findings that even robust LLMs like `GPT-4` struggle with task compositionality. The performance on hard examples, reflecting challenging multiple subgoals settings, could be more crucial than average metrics.

**Long-Range Interaction.** We analyze their progress rate over interaction steps (Figure 4). Models like `GPT-4`, `Claude2` show consistent progress over 30 steps in Alfworld and PDDL tasks. However, in WebArena and Tool tasks, their performance peaks early and then stagnates. This may be due to that later stages for these tasks are challenging. Open-weight models, except `Llama3-70b` and `Deepseek-67b`, peak early and generally stop progressing after about 6 steps, likely struggling with the increased complexity of long-range interactions and extended context requirements.

**Sub-skill Analysis.** We aim to assess LLMs across several facets: *memory* that measures incorporating long-range information in context, *planning* that assesses decomposing complex goals into manageable sub-goals, *world modeling* which tests knowledge necessary for task completion, *self-reflection* that captures the ability to use environmental feedback, *grounding* that focuses on competency in generating valid actions, and *spatial navigation* that represents efficiency in moving to a target location. We develop a sub-skill scoring system based on Table 11 As depicted in Figure 5, `GPT-4` surpasses all other LLMs across all sub-skills.

**Exploration Behavior.** Analysis on the exploration behavior are available in Appendix E.

# 5 Visualization Panel for LLM Agent Analysis: A Case Study

We use Weights&Bias for our visualization panel with task boards for individual task analysis (§2 and §4). As shown in Appendix Figure 7, for `GPT-4`, we have `GPT-4` as *Current Run*, and 6 other models as baselines for comparison. We first look at the summary board, showing `GPT-4` outperforms all baselines by a large margin in terms of overall metrics. Also, `GPT-4` demonstrates high capability score on all 6 subskills. From the radar plot "summary/all results" we can see that `GPT-4` performs the worst on Jericho and WebArena, and we can check their respective task board for more information. In the Jericho task board, it is evident that although the performance metrics of `GPT-4` are relatively low, it still outperforms other baselines. However, the performance notably declines for challenging examples, as indicated in the "jericho/progress rate w.r.t difficulty" bar plot. To further investigate, we can examine the trajectory of several failed cases in the "jericho/predictions" table. For instance, in the "zenon" sub-task, the agent successfully unlocks the cell door but fails to distract the guards, resulting in an inability to escape. This failure can be attributed to the limited exploration ability of the agent, as it should have explored the available gadgets in the room to distract the guards.

# 6 Related Work

**LLM as Agent** Traditional Reinforcement Learning offers general decision-making solutions but struggles with sample efficiency and generalization (Pourchot and Sigaud, 2019). In contrast, the emergent reasoning and instruction-following abilities of LLMs (Wei et al., 2022) enable them to excel as agents (Yao et al., 2023; Richards, 2023; Wang et al., 2023a). The primary method for employing LLMs as agents involves prompting them with task instructions and environmental context to generate actionable responses (Richards, 2023; Xie et al., 2023). Specialized training can further enhance their agentic capabilities (Xu et al., 2023c; Reed et al., 2022; Driess et al., 2023). We benchmark both general (OpenAI, 2023; Touvron et al., 2023; Chiang et al., 2023) and agent-specific LLMs (Xu et al., 2023c) to study their effectiveness as agents. Additionally, research explores various dimensions of agent abilities, including grounding goals to actions (Gu et al., 2022; Ahn et al., 2022), world modeling (LeCun, 2022), step-by-step planning (Song et al., 2023), and self-reflection (Madaan et al., 2023; Wang et al., 2023b). Evaluating these skills is essential to understand limitations of LLMs as agents.

**Evaluating LLM in Decision Making Problems** Several benchmarks and toolkits for LLM agents have been established, focusing on various tasks such as web-browsing, games, and tool use (Yao et al., 2022; Zhou et al., 2023; Shridhar et al., 2021; Qin et al., 2023a; Wang et al., 2023a; Ye et al., 2024; Kinniment et al., 2023). A few other benchmarks provide a proof-of-concept study on specific LLM features, with Wang et al. (2023b) focusing on model interaction ability, and Liu et al. (2023b) examining agent structures. Recent works by Liu et al. (2023a); Wu et al. (2023); Mialon et al. (2023) present a generalist challenge for LLM agents, please refer to Table 1 for a comparison. Note that recent progress in multimodal LLMs has spurred research into multimodal LLM agents (Zheng et al., 2024; Yang et al., 2023). Our study focuses exclusively on text-based environments to assess LLM agent abilities via textual reasoning and actions in-depth.

## 7 Conclusion

In this work, we introduce AGENTBOARD as a benchmark for evaluating generalist LLM agents. In addition to being a benchmark, AGENTBOARD offers an open-source, analytical evaluation framework that facilitates easy customization, unified metrics, and comprehensive analysis from diverse aspects, in addition to an interactive visualization web panel. Such analytical evaluation is equipped with an interactive visualization web panel, allowing users to efficiently explore the evaluation and gain a deeper understanding of the agents of interest. Overall, AGENTBOARD aims to facilitate detailed evaluation and understanding of LLM agents, driving further advancements in the field.

**Limitations:** Limitations of AGENTBOARD include reliance on human-annotated subgoals to calculate progress rate. Although using LLMs for annotation is considered, current models underperform on AGENT-BOARD tasks and cannot accurately generate subgoals. Additionally, AGENTBOARD evaluates agents mainly in simulated environments to maintain standardization. However, real-world benchmarking is crucial for practical applications but presents challenges such as variable ground truth labels and security risks. We will address them in future work.

## Acknowledgement

We thank Tao Yu, Shuyan Zhou for providing valuable comments on research questions and experimental design. We thank Yiheng Xu, Haiteng Zhao and Hongjin Su for early stage beta testing. Zhihao Zhu and Yaohui Jin are with the MoE Key Lab of Artificial Intelligence, Al Institute, Shanghai Jiao Tong University, and Zhihao Zhu is supported by Shanghai Municipal Science and Technology Major Project (2021SHZDZX0102) and the Fundamental Research Funds for the Central Universities. We thank wandb for free logging and backing the engine of AGENTBOARD.

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

# Appendix

## A    Author Contributions

**Code Implementation**    Chang Ma implemented the code base for AgentBoard framework. The code for different tasks is implemented by respective person in charge: Junlei Zhang (Alfworld, Scienceworld), Chang Ma (BabyAI, Jericho and PDDL), Zhihao Zhu (WebShop and WebArena), Cheng Yang (Tool-Query and Tool-Operation). The website was implemented by Zhihao Zhu and the visualization panel was implemented by Chang Ma. The code of Alfworld, ScienceWorld, PDDLGym, WebShop, WebArena and Mint sped up the implementation.

**Task Unification**    Junlei Zhang, Chang Ma, Cheng Yang, Zhihao Zhu implemented the tasks into environmental interaction format, provided labels for respective tasks, adapted the metrics, and verified the performances. Chang Ma, Junlei Zhang, Junxian He additionally verified the tasks to be unified.

**Paper writing**    Chang Ma and Junxian He finished introduction and methodology sections of the paper. Junlei Zhang and Chang Ma wrote the experiments section. Cheng Yang provided all the visualizations shown in the paper. Cheng Yang, Zhihao Zhu added results and analysis for their corresponding parts. Junxian He carefully reviewed and revised the paper and gave feedback for multiple rounds. Other authors help proofread and provide feedbacks.

**Experiments**    Chang Ma and Junlei Zhang co-lead the evaluation of the models. Zhihao Zhu conducted all the evaluations on web tasks. Cheng Yang conducted evaluation on tool tasks for several models and conducted experiments on analysis and visualization.

**Data Collection and Human Annotation**    Data for task examples and progress rate annotation for each task is collected and annotated with one person in charge, and verfied by at least two others: Junlei Zhang led data collection and annotation for ScienceWorld, Alfworld; Chang Ma led data collection and annotation for BabyAI, Jericho and PDDL; Zhihao Zhu led data collection and annotation for WebShop, WebArena and Sheet task in Tool-Operation; Cheng Yang led data collection and annotation for Tool-Query and Tool-Operation. Junlei Zhang led data validation for BabyAI and Tool-Query; Chang Ma led data validation for ScienceWorld and WebShop; Zhihao Zhu led data validation for Jericho, PDDL and Tool-Operation; Cheng Yang led data validation for ScienceWorld and WebArena.

Junxian He is the main advisor of this project.

## B    Limitations

While AGENTBOARD attempts to circumvent the current pitfalls of LLM benchmarks, some limitations remain:

**Human-dependent Annotation**    One limitation of AGENTBOARD is its reliance on human-annotated subgoals to measure the progress rate of LLM agents. The subjectivity of annotators calls for multiple verifications just to ensure the uniformity of the data. Also, as the complexity of tasks increases, the number of subgoals and the granularity required in annotation can grow significantly. This makes the process less scalable and more labor-intensive. One alternative is to use LLMs instead of humans to annotate these subgoals, but all LLMs currently underperform on AGENTBOARD tasks and cannot accurately generate subgoals. We look forward to better LLMs that could autonomize planning subgoal annotation.

**Benchmarking Real-World Problems**    The price to pay for a standardized and definitive agent benchmark is to evaluate agents in simulated environments. However, it's important to also benchmark on real-world problems for future applications. Currently, there are a few challenges with benchmarking on real-world problems that we hope to overcome in subsequent work: (1) Variable ground truth labels: most real-world environments are ever-changing, e.g., contents on a web page, and this could lead to changes in states and labels for LLM agents, creating difficulties for benchmarking. (2) Security measures: we benchmark not only on obtaining information from environments but also on operating within and altering these environments. In real-world scenarios, it could be dangerous to unleash an LLM agent, e.g., on the Internet, which could lead to harm or generate malicious information. Therefore, it is very important yet challenging to constrain the LLM agent's action space in the real world while still offering it the freedom to accomplish tasks.

## C  Ethics and Societal Impact

This paper presents work whose goal is to advance the field of Machine Learning. Potential societal implications include we allow LLM agents to access online API in tool-operation evaluation. LLM agents could access and edit online information including Google Sheet and To-do list. However, we made sure that no personal information is leaked and no generated content is distributed online during the evaluation process.

## D  Confidence Interval of LLM Agents Evaluation

In table 6 and 7, we report the confidence interval of representative models. Notably, when comparing proprietary models to open-source models, the former often exhibit larger deviations in their outputs. This can be primarily attributed to the decoding process of proprietary models, which typically involves additional post-processing steps. Open-source models demonstrate smaller deviations as we have chosen to use smaller temperatures to ensure the reproducibility of the experiments.

| Models | ALF | SW | BA | JC | PL | WS | WA | TQ | TO | Avg |
|--------|-----|-----|-----|-----|-----|-----|-----|-----|-----|-----|
| GPT-4 | 69.1±11.6 | 78.3±3.0 | 65.7±7.5 | 45.4±12.9 | 79.7±2.7 | 77.1±1.4 | 42.5±4.4 | 85.6±0.7 | 79.1±6.2 | 70.3±4.7 |
| GPT-3.5 | 33.6±4.3 | 17.5±13.3 | 47.0±5.5 | 18.4±9.7 | 26.0±2.9 | 78.1±4.8 | 21.6±5.3 | 65.0±3.8 | 39.1±2.0 | 38.5±3.1 |
| Mistral-7b | 9.7±0.1 | 15.6±0.2 | 20.1±0.1 | 11.2±0.4 | 4.5±0.3 | 67.4±0.6 | 13.6±0.4 | 36.0±0.1 | 25.2±1.4 | 22.6±0.2 |
| Llama-13b | 8.3±1.6 | 1.4±0.3 | 15.8±4.0 | 7.0±3.3 | 4.8±1.0 | 60.9±2.3 | 7.9±0.0 | 34.5±2.6 | 31.0±2.9 | 19.1±1.9 |

Table 6: Confidence Interval of Progress Rate Metrics

| Models | ALF | SW | BA | JC | PL | WS | WA | TQ | TO | Avg |
|--------|-----|-----|-----|-----|-----|-----|-----|-----|-----|-----|
| GPT-4 | 46.3±20.3 | 50.4±9.6 | 52.1±5.8 | 26.7±10.4 | 59.5±3.9 | 39.9±0.8 | 20.1±7.0 | 70.0±1.7 | 54.2±12.3 | 47.8±5.4 |
| GPT-3.5 | 13.5±5.2 | 7.5±9.9 | 36.3±3.2 | 1.7±2.9 | 6.1±1.9 | 31.7±4.2 | 5.3±1.7 | 41.1±3.5 | 10.8±2.9 | 16.9±2.5 |
| Mistral-7b | 0.0±0.0 | 2.2±0.0 | 14.2±0.1 | 0.0±0.0 | 0.0±0.0 | 14.5±0.5 | 1.7±0.4 | 0.00±0.0 | 0.00±0/0 | 3.6±0.1 |
| Llama-13b | 0.0±0.0 | 0.0±0.0 | 5.6±1.0 | 0.0±0.0 | 0.0±0.0 | 10.0±0.7 | 2.0±0.0 | 0.0±0.0 | 0.0±0.0 | 0.6±0.2 |

Table 7: Confidence Interval of Success Rate Metrics

## E  Exploration Behavior Analysis

We examine the exploration behavior of models in various environments, as illustrated in Table 8. The ability of agents to explore plays a significant role in their performance in partially-observable environments, as diverse exploration trajectories enable agents to acquire all the necessary information. We compare the number of locations explored by models, including rooms in BabyAI, containers in AlfWorld, and places in Jericho. This metric reflects the models' exploration capabilities. Most models are unable to explore the minimum number of locations necessary to complete the goal. `GPT-3.5-Turbo` demonstrates performance comparable to `GPT-4` in this score. Among the open-weight models, `Llama2-70b` and `CodeLlama-34b` show similar performance and both outperform `Vicuna-13b-16k`, consistent with progress rate and success rate. Detailed analysis on exploration behavior is not currently implemented in our AGENTBOARD framework since it is feasible only for some environments.

| Tasks | Minimum | GPT-4 | GPT-3.5-Turbo | Llama2-70b | CodeLlama-34b | Vicuna-13b-16k |
|-------|---------|-------|---------------|------------|---------------|----------------|
| Babyai - UnlocktoUnlock | 3 | 1 | 1.25 | 1 | 1.25 | 1 |
| Babyai - FindObjs5 | 3 | 2 | 3.5 | 2 | 2 | 1 |
| Babyai - Keycorridor | 3 | 3 | 2.5 | 1.5 | 1.75 | 1.25 |
| Alfworld | 3 | 5.625 | 5.125 | 1.75 | 0.125 | 1 |
| Jericho - Zork1 | 5 | 5 | 5 | 1 | 5 | 1 |
| Jericho - Zork2 | 6 | 6 | 2 | 1 | 3 | 3 |
| Jericho - Zork3 | 11 | 6 | 4 | 3 | 2 | 1 |

Table 8: Comparison of the number of locations(room in babyai, containers in alfworld, and places in Jericho) explored by models. The minimum column states the least number of locations need to explore on average in order to finish the tasks.

## F  Ablation Study of Agent Framework

In this part we discuss alternations of framework design for testing LLM agents. Our principal goal is to test the **basic** agentic abilities of LLM, which calls for a simplistic framework to avoid introducing confounders. Currently there are many modular-based agent frameworks (Wang et al., 2023a; Hong et al., 2023; Wu et al.,

2024) that could improve performances of agents. However, these frameworks often involve intricate design and not applicable to all LLMs. Therefore, we choose Act (Yao et al., 2023) as our framework, which requires minimal design and is applicable to most instruction-following LLMs.

| Task Name | *Act* | *ReAct* |
|---|---|---|
| AlfWorld | 35.6/17.2 | 37.9/ 8.2 |
| PDDL | 25.0/ 5.0 | 17.1/ 3.3 |
| Tool-Query | 69.4/45.0 | 70.0/50.0 |
| Tool-Operation | 37.2/ 7.5 | 54.8/10.0 |

Table 9: Comparison of Act and ReAct framework using GPT-3.5-Turbo.

One popular alternative for our framework is to use ReAct (Yao et al., 2023) rather than Act, which uses interleaved thoughts in addition to actions to boost planning. However, our experiments on GPT-3.5-Turbo show inconsistent improvement of ReAct over performance of Act, as shown in Table 9. We hypothesized that this is due to we test long-term interactions of LLM Agents up to 30 interactions, and adding thoughts would pressure context length, thus leading to performance drop. Therefore, we use Act rather than ReAct in our benchmark.

Another major alternation is how to handle out of context length prompts during agent prompting. Here we provide ablation study on alternatives to our *sliding window* approach:

- *Sliding Window* (Current Approach): We keep record of most recent interaction history within the prompt.
- *Cutoff*: This approach has been used in AgentBench (Liu et al., 2023a). It removes the sliding window and stops the interaction when the prompt (including history) overflows the maximum context length.
- *Summary*: This approach has been used in Xu et al. (2023a); Chase (2022). It uses the LLM to generate a summary of history to replace the interaction history when the prompt overflows maximum context length.

'

We benchmark on `GPT-3.5-Turbo` with only 4k context length and `Mistral` with 32k context length. The results are shown in Table 10. First, the memory component design barely affects models with long context length (`Mistral`), while greatly affects `GPT-3.5-Turbo`. The Summary method depends on the summarization ability of the LLM, therefore its performance varies between tasks. The cutoff method generally performs worse than sliding windows, though it is more stable than summary.

This shows that sliding window enables LLMs to make use of its limited context length. Its simple design also avoids introducing confounders into our benchmark, e.g. summarization abilities of LLM.

# G   Sub-Skill Table

Table 11 shows the criteria for sub-skill scoring and sub-skill scores for each task in AGENTBOARD.

# H   Visualization Panel

The visualization panel supported by AGENTBOARD is shown in Figure 6. We provide a detailed explanation of panel features and usage tutorial in WandB blog[3].

We provide a case study in Figure 7 to show example usage of AGENTBOARD.

# I   Details of Evaluated LLMs

## I.1   Evaluation Setup

We use greedy decoding strategy and set temperature to zero for better replicacy, and all LLMs are implemented with vLLM (Kwon et al., 2023) architecture, which has $10\times$ acceleration over huggingface inference. During prompting, we keep the most recent interaction histories within the maximum context length of the model. For models with different versions of checkpoints, we choose the version with best instruction following ability, with chat SFT and alignment. The following are the specific models we assess in the experiments.

---

[3]This blog will be released after the review period

|  | ALF | SW | BA | JC | PL | WS | WA | TQ | TO | Avg. |
|---|---|---|---|---|---|---|---|---|---|---|
| GPT-3.5 + sliding | 35.6/17.2 | 31.9/18.9 | 51.7/39.3 | 19.9/5.0 | 25.0/5.0 | 76.4/35.1 | 25.5/4.6 | 69.4/45.0 | 37.2/7.5 | **41.4/19.7** |
| GPT-3.5 + cutoff | 28.1/6.7 | 1.5/0.0 | 47.2/35.7 | 9.4/0.0 | 18.1/1.7 | 73.6/31.1 | 12.1/4.1 | 62.5/36.7 | 42.8/7.5 | 32.8/13.7 |
| GPT-3.5 + summary | 29.4/6.7 | 2.2/0.0 | 44.4/33.0 | 5.9/0.0 | 19.1/3.3 | 75.2/31.5 | 12.8/4.1 | 63.3/38.3 | 49.2/12.5 | 33.5/14.4 |
| Mistral-7b + sliding | 9.8/0.0 | 15.8/2.2 | 20.1/14.3 | 11.0/0.0 | 4.7/0.0 | 68.2/13.9 | 13.2/1.3 | 51.0/3.3 | 27.2/0.0 | **24.6/3.9** |
| Mistral-7b + cutoff | 9.7/0.0 | 15.4/2.2 | 20.1/14.3 | 11.7/0.0 | 3.5/0.0 | 70.3/14.7 | 9.9/1.6 | 49.7/1.7 | 25.7/0.0 | 24.0/3.8 |
| Mistral-7b + summary | 9.7/0.0 | 15.4/2.2 | 20.1/14.2 | 11.0/0.0 | 6.0/1.7 | 67.1/14.7 | 9.9/1.6 | 49.7/1.7 | 26.7/0.0 | 23.9/4.0 |

Table 10: Comparison of various memory approaches to handle long-context agent prompting.

|  | AlfWorld | ScienceWorld | BabyAI | Jericho | PDDL | WebShop | WebArena | Tool-Query | Tool-Operation |
|---|---|---|---|---|---|---|---|---|---|
| **Memory** |  |  |  |  |  |  |  |  |  |
| 1. Could finish tasks within 2k tokens
2. Could finish task within 4k tokens
3. Otherwise | 1 | 2 | 1 | 1 | 2 | 1 | 3 | 2 | 3 |
| **Planning** |  |  |  |  |  |  |  |  |  |
| 1. ≤ 3 subgoals on average
2. ≤ 5 subgoals on average
3. Otherwise | 1 | 2 | 2 | 3 | 3 | 2 | 3 | 2 | 2 |
| **World Modeling** |  |  |  |  |  |  |  |  |  |
| 1. Requires no additional knowledge other than instruction
2. Requires knowledge of the environment from exploration
3. Requires commonsense knowledge in addition to knowledge from environment | 3 | 3 | 2 | 3 | 1 | 1 | 3 | 1 | 1 |
| **Self-Reflection** |  |  |  |  |  |  |  |  |  |
| 1. Detailed feedback and error message with instruction for the next step.
2. Not very detailed feedback and error message
3. No error message, e.g. "no change in state" | 3 | 2 | 2 | 1 | 3 | 2 | 2 | 1 | 1 |
| **Grounding** |  |  |  |  |  |  |  |  |  |
| 1. No specific action format is required, could recognize similar actions
2. Action format is required
3. Action format hard to follow | 2 | 3 | 2 | 1 | 3 | 3 | 3 | 3 | 3 |
| **Spatial Navigation** |  |  |  |  |  |  |  |  |  |
| 0. No spatial navigation
1. 2D navigation | 1 | 1 | 1 | 1 | 0 | 0 | 1 | 0 | 0 |

Table 11: The sub-skill scores associated with each task in AGENTBOARD.

| Model Name | Model Code/API |
|---|---|
| GPT-4 (OpenAI, 2023) | Azure api: `gpt-4` (version: 2023-05-15) |
| GPT-3.5-Turbo (OpenAI, 2022) | Azure api: `gpt-35-turbo` |
| GPT-3.5-Turbo-16k (OpenAI, 2022) | Azure api:`gpt-35-turbo-16k` |
| Claude2 (Anthropic, 2023) | Anthropic api: `claude-2` (version: 2023-06-01) |
| Claude3-Haiku (Anthropic, 2023) | Anthropic api: `claude-30-haku` (version: 2024-03-07) |
| Gemini-1.5-Flash (Anthropic, 2023) | Google api: `gemini-1.5-flash` |
| Text-Davinci-003 (Ouyang et al., 2022) | Azure api: `text-davinci-003` |
| Llama3-8b (Touvron et al., 2023) | `meta-llama/Meta-Llama-3-8B-Instruct` |
| Llama3-70b (Touvron et al., 2023) | `meta-llama/Meta-Llama-3-70B-Instruct` |
| Mistral-7b (Jiang et al., 2023) | `mistralai/Mistral-7B-v0.1` |
| CodeLlama-13b (Roziere et al., 2023) | `codellama/CodeLlama-13b-Instruct-hf` |
| CodeLlama-34b (Roziere et al., 2023) | `codellama/CodeLlama-34b-Instruct-hf` |
| Llama2-13b (Touvron et al., 2023) | `meta-llama/CodeLlama13b-chat-hf` |
| Llama2-70b (Touvron et al., 2023) | `meta-llama/Llama-2-70b-chat-hf` |
| Vicuna-13b-16k (Chiang et al., 2023) | `lmsys/vicuna-13b-v1.5-16k` |
| Lemur-70b (Xu et al., 2023c) | `OpenLemur/lemur-70b-chat-v1` |
| DeepSeek-67b (DeepSeek-AI et al., 2024) | `deepseek-ai/deepseek-llm-67b-chat` |

Table 12: Model code/API of our evaluated models.

## I.2 Details of Models

We list our evaluated models in Table 12.

## J Data Quality Control

To ensure the quality of labeled sub-goals, we conducted three rounds of data verification for each labeled sub-goal. We developed an interactive interface through which inspectors complete tasks and observe the reward scores obtained at each step. If the inspector deems the reward score assigned during interaction with an environment to be unreasonable, additional annotators will engage in a discussion to determine if modifications to the labeled sub-goals are necessary.

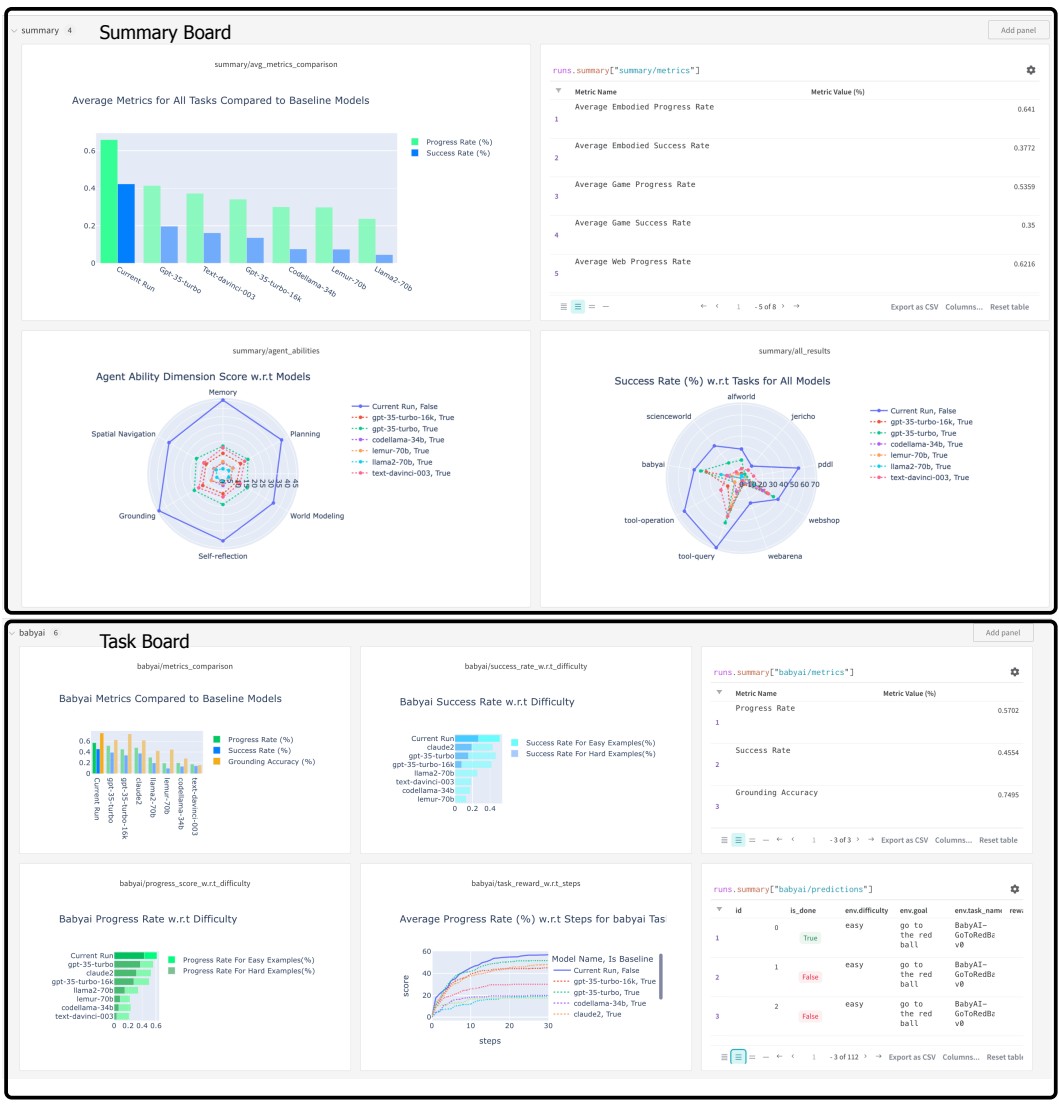

Figure 6: Visualization Panel based on WandB, composed of a summary board with all metrics and a task board for each task.

| Task | Proportion of environments with annotation errors |
|---|---|
| AlfWorld | 10.0% |
| ScienceWorld | 0% |
| Babyai | 4.2% |
| Jericho | 25% |
| PDDL | 5% |
| WebShop | 0% |
| WebArena | 0% |
| Tool-Query | 0% |
| Tool-Operation | 0% |

Table 13: The proportion of environments with annotation errors in the second round of data checking. Environments identified with errors are subsequently analyzed to determine the underlying causes, and any environments exhibiting similar errors are amended collectively.

The first round of verification is a self-check. Each annotator is required to carefully review the labeled tasks in every environment they are responsible for. The second round involves a sampled inspection by two annotators for each task. They examine a sample of 5-10 items from different sub-tasks within the task and document the

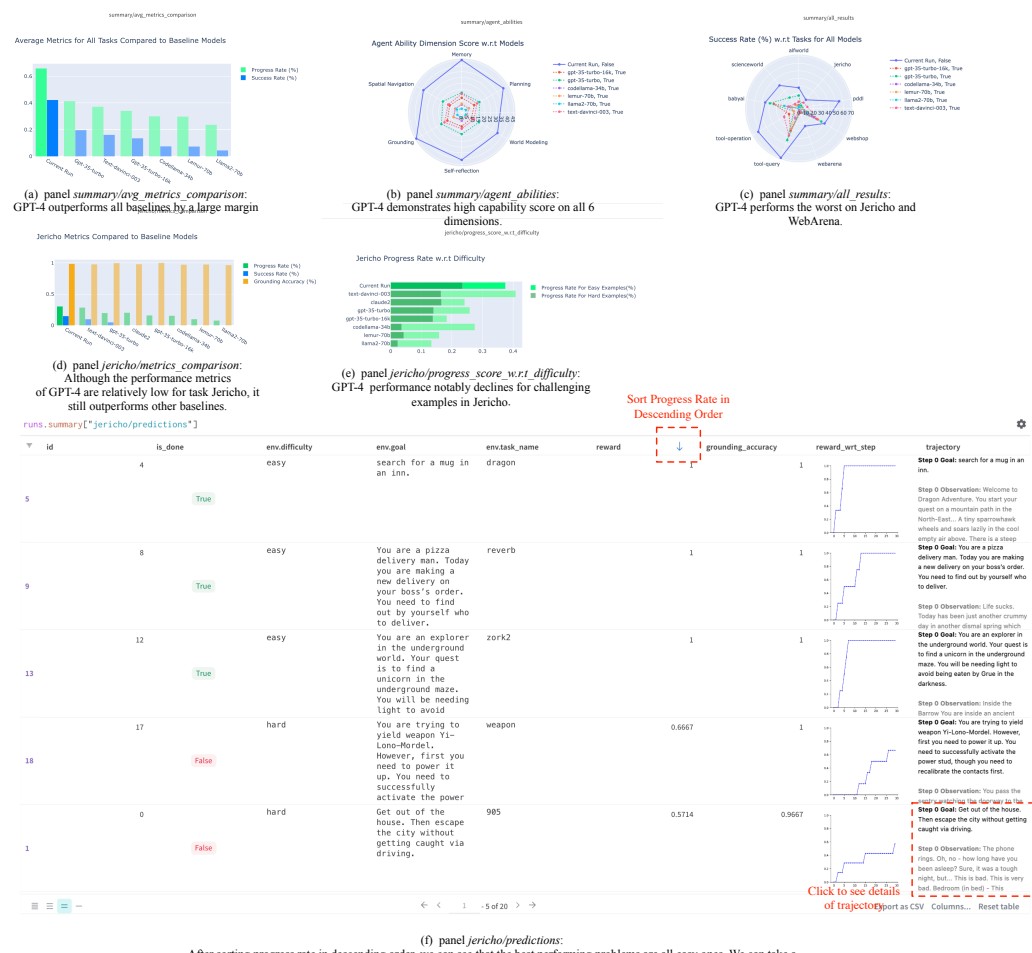

Figure 7: A case study for GPT-4 based on Panels from AGENTBOARD.

Table 14: Statistics of 9 environments in AGENTBOARD. "subgoal" and "match" means 2 different implementations of progress rate $r^{\text{subgoal}}$ and $r^{\text{match}}$ respectively. [†]Note that Sheet Environments in Tool-Operation are evaluated with $r^{\text{match}}$, while other sub-tasks are evaluated with $r^{\text{subgoal}}$. [‡]For tasks without subgoal label, we state the average number of constraints to satisfy in the goal state, which is essentially the complexity of the problems. For context length, we report the number of tokens generated with Llama2 tokenizer. [†] We divide problems into hard/easy based on the number of subgoals – problems with a larger number of subgoals than cutoff are viewed as hard.

| | Embodied AI | | | Game | | Web | | Tool | |
|---|---|---|---|---|---|---|---|---|---|
| | ALF | SW | BA | JC | PL | WS | WA | TQ | TO |
| # Environment | 134 | 90 | 112 | 20 | 60 | 251 | 245 | 60 | 40 |
| # Turns | 6 | 15 | 10 | 20 | 20 | 3 | 25 | 5 | 6 |
| Action Space | 13 | 21 | 8 | 150 | 8 | 2 | 12 | 15 | 16 |
| # Avg. Subgoals[‡] | 3 | 5 | 4 | 6 | 6 | 4 | 6 | 5 | 5 |
| Hard/Easy Cutoff[†] | 3 | 3 | 3 | 4 | 6 | 1 | 4 | 4 | 4 |
| Context Length | 900 | 2800 | 1800 | 1500 | 2700 | 1200 | 15000 | 2100 | 4300 |
| Progress Rate | subgoal | subgoal | subgoal | subgoal | match | match | match | subgoal | subgoal/match [†] |
| Success Rate | (Progress Rate == 1) | | | | | | | | |

proportion of issues identified, as presented in Table 13. The third round is conducted by an annotator who is well-acquainted with the various tasks, who then performs a sampled review of all tasks.

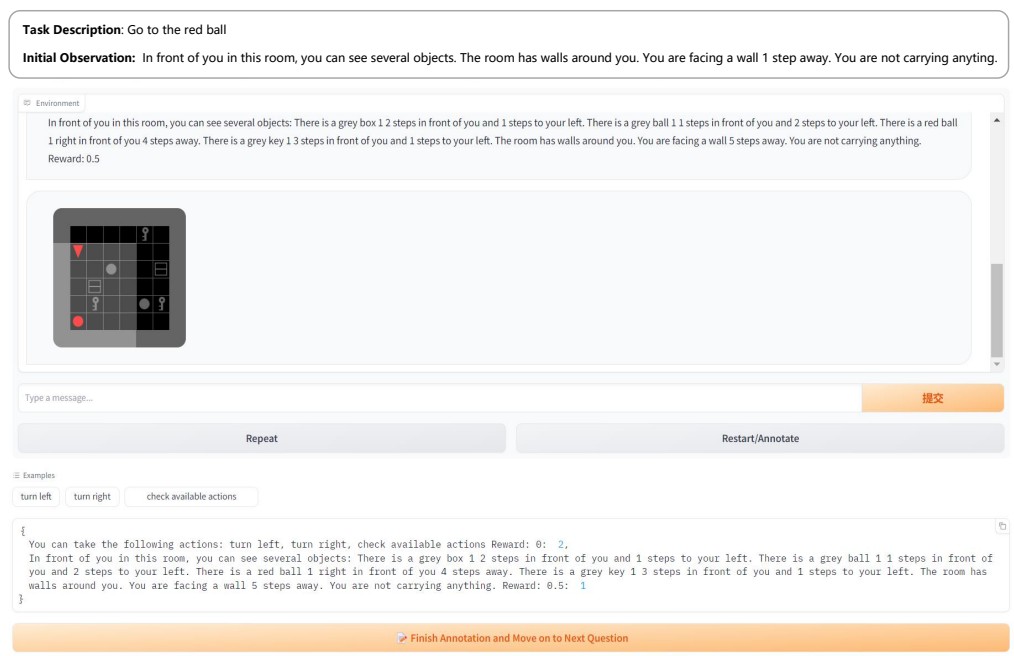

Figure 8: An illustration of our sub-goal checking interface. We develop an interactive interface for annotators to checking sub-goals. Firstly, annotators play and pass the game with the interface. The reward score for each step will be given based on the labeled score. If the annotators are dissatisfied with the reward, annotators will record them and the corresponded environment will be discussed by more annotators and annotated again."

# K    Details of Environments

## K.1    Details of Embodied Environments

**AlfWorld (ALF) (Shridhar et al., 2021)**    are Household tasks that require models to explore rooms and use commonsense reasoning to perform tasks, Within AGENTBOARD, we evaluate a model's ability to perform tasks in physical household settings, such as "put a pencil on the desk". AlfWorld is categorized into six types, comprising a total of 134 environments.

**ScienceWorld (SW) (Wang et al., 2022)**    is a complex interactive text environment that poses a significant challenge to agents' scientific commonsense. This environment requires agents to navigate through 8 distinct functional rooms (e.g., workshop, kitchen) and utilize the tools to complete tasks such as "measure the melting point of the orange juice". To address these issues, we re-annotate subgoals to calculate $r_t^{subgoal}$, where Specifically, we incorporate necessary observations as part of the subgoals. the rewards for these subgoals are uniform and distributed evenly throughout the task. To ensure that our annotated subgoals are necessary for achieving final goals, we restrict the use of tools and designated task completion rooms in the task descriptions. We show more details of we annotated subgoals in the Appendix L.3

**BabyAI (BA) (Chevalier-Boisvert et al., 2019)**    is an interactive environment where agents navigate and manipulate objects in a 20x20 grid space. The agent can only see objects within a limited sight and cannot perceive objects in remote rooms. The original implementation represents observations as images and only allows for tensor-based low-level actions such as "0: move left". "1: move right", and "2: move forward". To enable text-based input and output for LLM agents, We modified it by mapping the original actions to a textual action space and providing textual descriptions of visual observations, as shown in Table 2. For each step, the environment returns a text description of the current observation, such as "There is a red ball 1 step to your right and 1 step ahead of you. There is a wall 2 steps ahead." We also introduced high-level actions, such as "go to red ball 1" and "toggle and go through green locked door 1", to expand the action space and enrich the

semantic complexity of the environment. Additionally, we implemented a new subgoal-based progress rate for the environments to increase the density of rewards compared to the original reward scores. Unlike the previous reward score in BabyAI which awards a point only after a new object is found or pickup, requiring many steps to see progress in reward score, our new approach increases density of the rewards, requiring fewer steps to achieve them. We re-annotated subgoals and calculate with the equation of $r_t^{\text{subgoal}}$. Subgoals are re-annotated to update the progress rate whenever the agent makes progress, such as navigating to another room, finding a red ball, and picking it up in the problem "pickup a red ball".

## K.2  Details of Game Environments

Evaluating LLM agents as strategic game playing agents demands strong planning ability of agents. We choose three tasks that are all demanding in planning and making strategies.

**Jericho (JC) (Hausknecht et al., 2020)**  is a collection of text-based game environments that evaluate agents to perform adventures in fictional worlds. This task is unique in that it requires strong world modeling ability as agents could only gain information about the magic world through exploration and interaction. For example, for the task that requires the agent to perform actions with magic, it cannot reason with pre-trained commonsense knowledge and must perform exploration to understand the rules of the magic world. The original games are quite long (need 50-300 steps to finish), which is not suitable for LLM agents with fixed context length. To solve this issue, we rewrite the goal of each adventure to restrict the games to be finished within 15 subgoals. For example, *zork1* game requires the player to enter a dungeon and explore the dungeon to find a bar. We rewrite the goal as "You need to find your way into a secret passage where the entrance is in the living room of the house." and the agent only needs to find the entrance to the dungeon, which can be finished in 8 steps. We use the $r_t^{\text{subgoal}}$ as progress rate metrics, and we meticulously annotate the subgoals for each problem. Each subgoal characterize that the agent has solved a small problem, e.g. "find the entrance to the house" → "enter the house" → "find the living room" → "discover a trap door" → "find the entrance to dungeon".

**PDDL (PL) (Vallati et al., 2015),**  short for Planning Domain Definition Language, is a set of strategic games defined with PDDL symbolic language. We selected 4 representative game domains, *Gripper, Barman, Blocksworld, Tyreworld* to benchmark LLM agents in diverse scenarios, where the agent needs to move balls across rooms, make cocktails, rearrange blocks and pump up and install new tyres to cars. This task is difficult as it requires multiple rounds of planned actions to finish a single subgoal and agents need to plan strategically to avoid repetitive steps. For example, in *Barman*, the player is given a menu, and is required to make a few cocktails with a few containers and ingredients. The agent could use a strategy of trying to use different containers each time to avoid repetitive cleaning and save steps. While the commonly-used environment implementation (Silver and Chitnis, 2020) requires the agent to interact with an environment with PDDL expressions, e.g. `clean-shaker(hand1, hand2, shaker)` and provides observations as set of predicates `ontable(shaker1)` $\wedge$ `empty(shaker1)`. we write parser rules to offer a text-based observation to agents that allows LLMs to interact with natural language to be consistent with other tasks. e.g."Shaker1 is on the table. Shaker1 is empty" and enable the agents to interact with the environment with simple text commands, e.g. "clean-shaker shaker1 with hand1 while hand2 is empty." We curate 10-20 problems for each of the four domains by ourselves, ensuring the problems are multi-round and diverse. We use the $r_t^{\text{match}}$ as progress rate metric, where the matching score compares the similarity between the properties of current state and the goal state. e.g. for the goal state "Block a is on block b. Block b is on the table", if at current state "Block a is on the table. Block b is on the table", then the matching score is 0.5. The agent will receive a 100% progress rate only if all conditions of the goal state are satisfied.

## K.3  Details of Web-based Environments

Evaluating LLM's capability as a generalist agent in web-based scenarios has become pivotal (Shi et al., 2017a; Deng et al., 2023). Web agent is expected to navigate the network efficiently and perform diverse tasks amidst highly dynamic, intricate, and multi-turn interactions. Based on the task categorization, we've pinpointed two tasks of high recognition and quality: the specific network task, WebShop (Yao et al., 2022), and the general network task, WebArena (Zhou et al., 2023). The latter permits unrestricted access to any supported webpage.

**WebShop (WS) (Yao et al., 2022)**  is a network-based simulation environment for e-commerce experiences, featuring a website with 1.18 million actual products, each with distinct labels and attributes. In this environment, the agent is allowed to interact with the system through 'search[QUERY]' or 'click[ELEMENT]' actions to purchase products matching the instructions. This process necessitates that the model possesses reasoning and grounding abilities. Based on the original implementation method (Yao et al., 2022; Shinn et al., 2023), we have improved the error feedback, including refining the observation for exceeding page limits and interacting with wrong objects. These enhancements contribute to the effective operation of the entire environment and the rationality of multi-step reasoning processes. As there are no sub-goals in the environment, to obtain a

continuous progress rate, we expanded the calculation rules from (Yao et al., 2022), calculating the score at different web pages (stages). To measure the distance of the current state to the final goal as the progress rate, we expanded the product scoring rules from Yao et al. (2022) to derive the score at different web pages. Please refer to Appendix L.7 for details.

**WebArena (WA) (Zhou et al., 2023)** is a real web environment containing four applications: online shopping, discussion forums, collaborative development, and business content management. It supports 11 different web browsing actions. such as click (element), new tab, goto (URL), etc., and offers additional tools like maps and wikis. The observation space consists of structured web content (the accessibility tree [4]). Completing tasks in this highly realistic environment requires the agent to possess strong memory, high-level planning, common sense, and reasoning abilities. Compared to other datasets (Deng et al., 2023; Shi et al., 2017b), WebArena offers multi-round and continuous web browsing interaction simulation. We filtered 245 instances from the original dataset for two main sub-tasks: Site Navigation and Contact & Config, each annotated with the target URLs or required content. To obtain the progress rate, we revised the existing method for calculating the final score (Zhou et al., 2023) and continuously computed the progress rate at each step, fusing the URL matching score with the content matching score, derived from the current URL and target URL, with the content matching score calculated based on the detected required content, as detailed in Appendix L.8.

### K.4 Details of Tool Environments

In AGENTBOARD, a tool contains a variety of functions, accessed by agents via function calling. These functions are the actions that LLM agents can take in tool environments. Drawing upon open datasets and APIs, we have developed a suite of five distinct tools, each encapsulated in its own environment. Tool Environments are categorized into two groups: Tool-Query Environments and Tool-Operation Environments, representing two general usage scenarios. Tool-Query Environments include Weather Environment, Movie Environment and Academia Environment. Tool-Operation Environments include Todo Environment and Sheet Environment.

#### K.4.1 Tool-Query Environments

**Weather Environment** Weather Environment enables LLM agents to use the weather tool to retrieve past, present and future weather data, encompassing temperature, precipitation and air quality across various locales. We use Python codes to integrate Open-Meteo API[5], implement the requisite functions and subsequently develop a weather tool.

**Movie Environments** Movie Environment grants LLM agents to use the movie tool to access cinematic data, encompassing film details, personnel and production companies. We incorporate the API and data from The Movie Database[6], implement the necessary functions, and thus establish the movie tool.

**Academia Environment** Academia Environment equips LLM agents the academia tool to query information related to computer science research, including academic papers and author information. In its development, we harness data from the Citation Network Dataset[7], craft the relevant functions, and subsequently construct the academia tool.

#### K.4.2 Tool-Operation Environments

**Todo Environment** Todo Environment facilitates LLM agents in querying and amending personal agenda data through the todo tool. We implement the todo tool based on the Todoist API[8].

**Sheet Environment** Sheet Environment allows LLM agents to use the sheet tool to access and modify spreadsheet data. We build our sheet tool upon the Google Sheets API[9].

---

[4]https://developer.mozilla.org/en-US/docs/Glossary/Accessibility_tree

[5]https://open-meteo.com/

[6]https://www.themoviedb.org/

[7]https://www.aminer.org/citation

[8]https://todoist.com/

[9]https://www.google.com/sheets/about/

# L  Details of Progress Rate Metrics

## L.1  Explanation and Adaptations for "Unique" Subgoal Sequence

We manually edit problems for a simpler progress rate calculation setup where each final goal aligns with a unique subgoal sequence. Here we emphasize two attributes of our adaptation.

**Agentboard allows for multi-trajectory inference paths:**  Note that a single sequence of subgoals is not equivalent to a single inference path trajectory – in fact, our progress rate can be applied to tasks with multiple inference paths. We only restrict examples that have multiple different sets of subgoals to fulfill the same final goal. However, a single set of subgoals allows for very diverse inference paths. For example, in BabyAI, a problem is "go to a red ball," where the agent needs to find a red ball behind it and then go to the red ball. It could either finish the task with the golden path "turn right -> go to red ball 1" or take detours while eventually finishing the task, such as "move forward -> go to grey box 1 -> move forward -> turn right -> go to red ball 1". Both trajectories fulfill subgoals "find a red ball" and "go to the red ball".

**Our adaptations for single-set subgoals only affect less than 5% of all problems:**  We only limit situations where a model has many diverse sets of subgoals, such as in ScienceWorld, where an agent could use either a stove or a blast furnace to vaporize liquid. We simplify it by specifying a single method and clarifying the goal "vaporize liquid" as "vaporize liquid with a stove." This kind of adaptation only applies to tasks in BabyAI and ScienceWorld, constituting less than 5% of all problems in AgentBoard. Most of the existing samples already satisfy this requirement, and our modifications do not change the task difficulty; they are solely for the purpose of annotation convenience. A full list of adaptations and their examples is detailed in Table 15.

| Task | Number of examples adapted for a single set of subgoals |
|---|---|
| AlfWorld | None |
| ScienceWorld | 36 examples (40%) |
| Babyai | 4 example (3%) |
| Jericho | None |
| PDDL | None |
| WebShop | None |
| WebArena | None |
| Tool-Query | None |
| Tool-Operation | None |

Table 15: The proportion of environments that require adaptations for the simple setup of a single set of subgoals.

## L.2  Alfworld

We identify and annotate the necessary subgoals using regular expressions. For instance, for the task "put a pencil on the desk", we annotate one necessary observation as "You pick up the pencil +". This expression would match observations like "You pick up the pencil 1". When the goal of an environment is achieved, the environment emits a task success flag. Specifically, for each environment, we labeled N-1 necessary subgoals as N-1 subgoals. The final success flag combined with the N-1 annotated subgoals constitutes the set of N subgoals.

## L.3  ScienceWorld

We compare our modified task descriptions and subgoals with the original ones in Table 16. In the original scheme, subgoals are categorized as "sequential subgoals" and "unordered and optional subgoals". For the former, achieving sequential subgoals alone is sufficient to receive full rewards (100 points). However, under the "unordered and optional subgoals", each completed task is only awarded low point (e.g. 1 point). These tasks are also important and necessary for accomplishing the given task. For instance, the "optional subgoals" outlined in Table 16, such as "be in the same location as the orange juice" and "have the substance alone in a single container" are necessary for the task and can help to evaluate a model's navigation and common sense abilities. It is inappropriate to assign such tasks a low score. Furthermore, the uneven distribution of "Sequential Subgoals" throughout the entire task process can lead to a disproportionately low score, which does not accurately reflect the model's progress. For example, if the model fails to complete the initial subgoals within the "Sequential Subgoals" category, which could be considerably distant from the start state, it can only achieve a very low score. This scoring method does not align with our motivation, which is to ensure that the progress rate adequately reflects the model's performance. Therefore, we have re-annotated the subgoals. Specifically, we label necessary observations as part of the subgoals.

In the original task descriptions, the possibility of multiple necessary tools being present in multiple rooms (e.g., a thermometer) creates multiple viable gold paths for task completion. Consequently, a single state may exhibit

| | Original Labels | Ours |
|---|---|---|
| **Task Description** | Your task is to freeze orange juice. First, focus on the substance. Then, take actions that will cause it to change its state of matter. | Your task is to freeze orange juice **in the kitchen**. **The objects you can use are a metal pot, a freezer, a thermometer, and a fridge.** Take actions that will cause it to change its state of matter to a solid state. **Finally, examine its altered state. You should wait and monitor the temperature of the water until it changes its state.** |
| **Subgoals** | **Sequential Subgoals:**
1. focus on substance
2. substance is in a liquid state
3. substance is in a solid state
**Unordered and Optional Subgoals:**
1. be in same location as orange juice
2. have substance alone in a single container
3. have object in cooler (fridge)
4. have object in cooler (freezer)
5. cool object by at least 5C | **Necessary Observations:**
1. You move to the kitchen.
2. The freezer is now open.
3. The fridge is now open.
4. the thermometer measures a temperature of (-?[0-9] \| -?[1-9][0-9] \| -?[1-9][0-9]2) degrees celsius
5. solid orange juice |

Table 16: Comparison between the original task description and subgoals of ScienceWorld and our labeled subgoals(Best viewed in color).

different progress levels across various gold paths. This disparity makes it challenging to assign a definitive progress rate to any given state. Therefore, in our task descriptions, we have restricted the locations and tools used for tasks to ensure the uniqueness of our goal paths and the necessity of observations. For the necessary observations, our initial observation is more close to the initial state and but still challenging.

we design an interactive UI framework (Figure 8). We ask one graduate student to interact with the environment and record the necessary observations to achieve the given goal. As a result, we revise the task descriptions to include sufficient information for achieving the subgoals and to ensure the gold path is unique. .

## L.4 BabyAI

The origin implementation of babyai provides a reward score. Different from the original reward, our progress rate is more dense and the agent does not need to accomplish many steps before getting a increase in score. Here we compare the difference between our progress rate and the original reward score, as shown in Table 17. We can see from this case that our progress rate better measures intermediate progress for agents.

The progress rate is labelled via an interactive UI framework (Figure 8). A graduate student interact with the environments and record the observations corresponding to subgoals needed to finish the problem.

| Problem: Unlock to Unlock | Steps with Score Increase (Original) | Steps with Score Increase (Ours) |
|---|---|---|
| 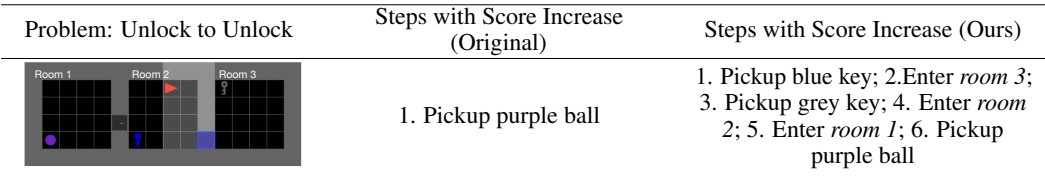 | 1. Pickup purple ball | 1. Pickup blue key; 2.Enter *room 3*; 3. Pickup grey key; 4. Enter *room 2*; 5. Enter *room 1*; 6. Pickup purple ball |

Table 17: Comparison between our progress rate for BabyAI and original reward score.

## L.5 Jericho

The original Jericho games are free-exploration text-based games, where the player is not given a tangent goal but allowed to explore around the environment as adventureres. For uniformity with other tasks, we first write a new goal for each problem, and we carefully select the goal so that the game could be accomplished within 15 subgoals. In contrast, the original environments requires around 50-300 interactions to get the maximum rewards. The annotation of goal and subgoals are also performed by a graduate student in the interactive UI framework.

## L.6 PDDL

In the PDDL environment, each state is discribed by a conjunction of properties $p_1 \wedge p_2, \ldots, \wedge p_m$, each property is a simple predicate describing the property of an object, e.g. "Block a is on the table". Given the goal state

$g_1 \wedge g_2, \ldots, \wedge g_n$ and any state $p_1 \wedge p_2, \ldots, \wedge p_m$, the matching score formula is defined as:

$$f = \frac{|\mathcal{G} \cap \mathcal{P}|}{|\mathcal{G}|}, \mathcal{G} = \{g_1, g_2, \ldots, g_n\}, \mathcal{P} = \{p_1, p_2, \ldots, p_m\} \tag{3}$$

The matching score is 1 if and only if the properties of goal state is satisfied in current state.

## L.7  WebShop

In the webshop environment, we expanded the product scoring rules from (Yao et al., 2022) to derive the score at different web pages. We can calculate the score of any product (the distance from the target product) using the original scoring formula as follows:

$$f = f_{\text{type}} \cdot \frac{|\mathcal{U}_{\text{att}} \cap \mathcal{Y}_{\text{att}}| + |\mathcal{U}_{\text{opt}} \cap \mathcal{Y}_{\text{opt}}| + \mathbf{1}[\text{y}_{\text{price}} \leq \text{u}_{\text{price}}]}{|\mathcal{U}_{\text{att}}| + |\mathcal{U}_{\text{opt}}| + 1}, \tag{4}$$

Each natural language instruction, denoted as $\text{u} \in \mathcal{U}$, encompasses a non-empty set of attributes, $\mathcal{U}_{\text{att}}$, a set of options, $\mathcal{U}_{\text{opt}}$, and a specified price, $\text{u}_{\text{price}}$. Meanwhile, $\mathcal{Y}$ represents the product chosen by the agent. The function $f_{\text{type}} = \text{TextMatch}(\bar{y}, \bar{y}^*)$ is based on text matching heuristics to assign low reward when $y$ and $y^*$ have similar attributes and options but are obviously different types of products.

Typically, completing a web shopping task involves three continuous stages: search, product selection, and finalizing the product style before placing an order. Therefore, to measure the distance between the current state and the target state, we primarily calculate scores for three pages (states): search result page, product description page, and order confirmation page. On the search result page, we calculate the score of each product on the page and take the highest score as the score for this page. On the product description page, we compute the highest score for the product under various options as the page score. On the order confirmation page, the score of the finally selected product is considered as the score for that page. In our method, the progress rate is the average of the scores from these three pages

## L.8  WebArena

In our method, we effectively utilize the annotation data, treating URLs as indicators of the web browsing trajectory and required contents as integral scoring points. The progress rate is formulated as follows:

$$r^{\text{match}} = \frac{n}{m+n}(r_d(r_q + r_p)) + \frac{m}{n+m}r_c \quad (n = 3; m = 0, 1, 2, \ldots) \tag{5}$$

Initially, we dissect the URL into its constituent elements: domain, query, and parameters by using `util.parse`. For domain verification, a binary value, $r_d$, is assigned, with a score of 1 indicating a correct domain match, and 0 otherwise. Subsequently, the matching score for the query, $r_q$, is determined through the application of the Longest Common Subsequence (LCS) algorithm, which assesses the similarity between the current and target queries based on their sequential nature. In contrast, the alignment between the current and target parameters is evaluated using the F1 score, denoted as $r_p$, which is particularly suited for unordered sets.

In parallel, the content matching score, $r_c$, emerges from the analysis of required content presence at each stage, calculated as the ratio of detected essential contents to the total required contents.

The overall progress rate integrates these two aspects, calculated as a weighted sum of the URL matching scores (incorporating domain, query, and parameter scores) and the content matching score. Here, $n$ represents the number of target URL components, and $m$ denotes the count of target required contents.

## L.9  Tool-Query

In Tool-Query Environments, we employ $r_t^{\text{subgoal}}$ as a metric to measure progress rate. Therefore, it is necessary to annotate subgoals for these envrionments. In Figure 9, we present an illustration of the process of subgoal annotation for Academia Environment. Specifically, when designing actions for these environments, we ensure that each action's functionality is indecomposable (i.e., the functionality and outcome of one action can not be achieved through other actions). This design choice results in a deterministic set of required golden actions to achieve our annotated goal. Furthermore, we ask human annotators to identify golden actions for each goal. Every output returned by executing golden actions is then processed as a subgoal.

## L.10  Tool-Operation

For Todo Environment, we adopt $r_t^{\text{subgoal}}$ as progress rate metric. Subgoals are annotated following the same process as Tool-Query Environments. In Sheet Environment, progress rate is assessed with $r_t^{\text{match}}$. Specifically,

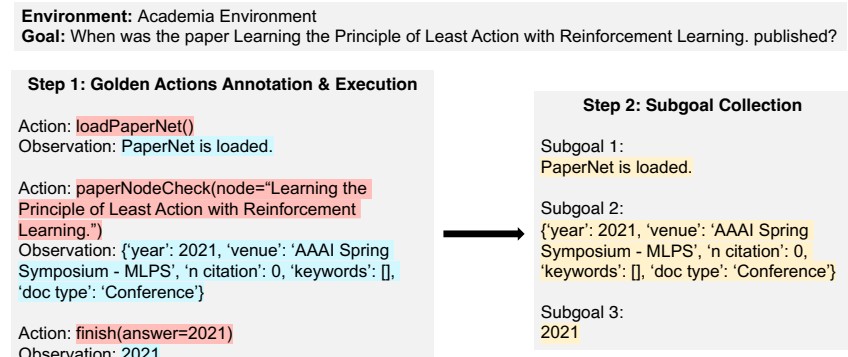

Figure 9: An illustration of the process of subgoal annotation for Academia Environment.

| Model | Device/API | Inference Architecture | Inference Speed | Total-time |
|---|---|---|---|---|
| GPT-4 | azure API | - | 1.5s/round | 5.5h |
| GPT-3.5-Turbo | azure API | - | 1s/round | 3h |
| DeepSpeed-67b | 8*V100 | vLLm | 5s/round | 18.5h |
| Llama2-70b | 8*V100 | vLLm | 8s/round | 28h |
| Llama2-70b | 4*A100 | vLLm | 4s/round | 13.5h |

Table 18: Inference Time Estimation

we first ask human annotators to annotate the golden spreadsheet for each goal. During the evaluation process, we calculate the matching score after each interaction round. The matching score is determined by the proportion of cells in current spreadsheet that align [10] with the golden spreadsheet.

## M  Runtime Estimation

The evaluation runtime for a language model depends on the device/API, model, and inference architecture used. In the case of open-source LLMs, the vllm inference speed is approximately 10 times faster than the huggingface pipeline. We show some time cost in Table 18.

## N  Prompt Details

As shown in Figure 10, we use a unified prompt template for different tasks in AGENTBOARD. Basically, a prompt consists of 5 parts. {System Prompt} represents the system prompt for the LLM, such as "You are a helpful AI agent". {Instruction} mainly consists of task descriptions and action definitions. {Examples} represents in-context learning examples. {Goal} is the current goal that needs to be accomplished, and {Trajectory} is the interaction history between the LLM agent and the environment.

For different tasks, the contents of these five parts are different. Prompt details for Embodied AI tasks are shown in Figure 11, 12 and 13. Prompt details for Game tasks are shown in Figure 14 and 15. Prompt details for Web tasks are shown in Figure 16 and 17. Prompt details for Tool tasks are shown in Figure 18 and 19, respectively.

---

Unified Prompt Template

{System Prompt}
{Instruction}
Here are examples:
{Examples}
{Goal}
{Trajectory}

---

[10] A cell is aligned if and only if its value is the same as the cell in the same position on the golden spreadsheet.

Figure 10: The unified prompt template in AGENTBOARD. {text} in blue font represents placeholders, which varies according to different tasks.

---

**Prompt Details for AlfWorld**

**System Prompt**
You are a helpful assistant. Generate your next step of action after Action. Action must not be empty. e.g. Action: put down cup.

- - - - - - - - - - - - - - - - - - - - - - - - - - - - - - - - - - - - - - - - - - - - - - - - - - -

**Instruction**
Your task is to interact with a virtual household simulator to accomplish a specific task. With each interaction, you will receive an observation. Your role is to decide on an action based on the observation. Please ensure that any objects ('obj') and receptacles ('recep') you mention in your response are present in the observation provided.
Here are the available actions you can take:
- take obj from recep
- put obj in/on recep
- open recep
- close recep
- toggle obj/recep
- clean obj using recep
- cool obj using recep
- heat obj using recep
- inventory
- examine recep/obj
- go to recep

- - - - - - - - - - - - - - - - - - - - - - - - - - - - - - - - - - - - - - - - - - - - - - - - - - -

**Examples**
Your task is to: look at statue under the desklamp.
You are in the middle of a room. Looking quickly around you, you see a coffeetable 1, a diningtable 1, a drawer 4, a drawer 3, a drawer 2, a drawer 1, a dresser 1, a garbagecan 1, a sidetable 2, a sidetable 1, and a sofa 1.
Action: go to dresser 1
Observation: On the dresser 1, you see a cellphone 3, a newspaper 2, a statue 1, and a television 1.
Action: take statue 1 from dresser 1
Observation: You pick up the statue 1 from the dresser 1.
Action: go to sidetable 1
Observation: On the sidetable 1, you see nothing.
Action: go to sidetable 2
Observation: On the sidetable 2, you see a desklamp 3, a newspaper 1, and a statue 2.
Action: use desklamp 3
Observation: You turn on the desklamp 3.

Figure 11: Prompt details for AlfWorld. The provided example is changed based on the type of the specific environment instance.

---

**Prompt Details for ScienceWorld**

**System Prompt**
You are a helpful agent that interacts with the virtual science school environment to solve the given task. Generate your next step of action after Action. Action must not be empty. e.g. Action: put down cup.

- - - - - - - - - - - - - - - - - - - - - - - - - - - - - - - - - - - - - - - - - - - - - - - - - - -

**Instruction**
You are an agent in a virtual science school environment, tasked to interact with various elements. Here are the commands you can use:

- **Manipulation**:
- open OBJ / close OBJ: Interact with a container.
- pick up OBJ: Add an object to your inventory.
- put down OBJ: Remove an object from your inventory.

- move OBJ to OBJ: Transfer an object.
- pour OBJ into OBJ: Pour a substance.
- dunk OBJ into OBJ: Immerse a container in a liquid.
- mix OBJ: Chemically combine contents.

- **Inspection**:
- look around: Survey your surroundings.
- look at OBJ: Examine an object closely.
- look in OBJ: Peek inside a container.
- read OBJ: Review written content.

- **Device Operations**:
- activate OBJ / deactivate OBJ: Toggle a device.
- use OBJ [on OBJ]: Utilize a device or item.

- **Movement**:
- go to LOC: Relocate.

- **Miscellaneous**:
- eat OBJ: Consume an edible item.
- flush OBJ: Activate a flushing mechanism.
- focus on OBJ: Direct attention to a particular object.
- wait [DURATION]: Pause for a specified period.

- **Information**:
- task: Recap your current objective.
- inventory: Display items you're carrying.

Where:
- OBJ: Object
- LOC: Location
- [DURATION]: Specified time

- - - - - - - - - - - - - - - - - - - - - - - - - - - - - - - - - - - - - - - - - - - - - - - - - - - -

**Examples**
Task Description: Your task is to boil water. For compounds without a boiling point, combusting the substance is also acceptable. First, focus on the substance. Then, take actions that will cause it to change its state of matter.

ACTION: look around
OBSERVATION: This room is called the hallway. In it, you see:
a picture
a substance called air
the agent
You also see:
A door to the green house (that is open)
A door to the living room (that is open)
A door to the art studio (that is open)
A door to the kitchen (that is open)
A door to the bedroom (that is open)
A door to the workshop (that is open)

ACTION: open door to kitchen
OBSERVATION: The door is already open.

ACTION: go to kitchen
OBSERVATION: You move to the kitchen.
...

Figure 12: Prompt Details for ScienceWorld.

**System Prompt**

You are an exploration master that wants to finish every goal you are given. Generate your next step of action after Action. Action must not be empty. e.g. Action: put down cup.

------------------------------------------------

**Instruction**

You are placed in a room and you need to accomplish the given goal with actions.
You can use the following actions:
- turn right
- turn left
- move forward
- go to `<obj> <id>`
- pick up `<obj> <id>`
- go through `<door> <id>`: `<door>` must be an open door.
- toggle and go through `<door> <id>`: `<door>` can be a closed door or a locked door. If you want to open a locked door, you need to carry a key that is of the same color as the locked door.
- toggle: there is a closed or locked door right in front of you and you can toggle it.

------------------------------------------------

**Examples**

Observation: In front of you in this room, you can see several objects: The room has walls around you. You are facing a wall 1 steps away. You are not carrying anything.
Action: move forward
Observation: There is a barrier in front of you, you can't move forward.
Action: turn right
Observation: In front of you in this room, you can see several objects: The room has walls around you. You are facing a wall 2 steps away. You are not carrying anything. Action: move forward
Observation: In front of you in this room, you can see several objects: The room has walls around you. You are facing a wall 1 steps away. You are not carrying anything.
Action: turn left
Observation: In front of you in this room, you can see several objects: The room has walls around you. You are facing a wall 1 steps away. You are not carrying anything.
Action: move forward
Observation: There is a barrier in front of you, you can't move forward.
Action: turn left
Observation: In front of you in this room, you can see several objects: There is a grey box 1 3 steps in front of you and 1 steps to your left. There is a grey ball 1 2 steps in front of you and 2 steps to your left. There is a red ball 1 right in front of you 5 steps away. There is a grey key 1 4 steps in front of you and 1 steps to your left. The room has walls around you. You are facing a wall 6 steps away. You are not carrying anything.
Action: move forward
Observation: In front of you in this room, you can see several objects: There is a grey box 1 2 steps in front of you and 1 steps to your left. There is a grey ball 1 1 steps in front of you and 2 steps to your left. There is a red ball 1 right in front of you 4 steps away. There is a grey key 1 3 steps in front of you and 1 steps to your left. The room has walls around you. You are facing a wall 5 steps away. You are not carrying anything.
...

Figure 13: Prompt details for BabyAI.

**System Prompt**

You are a game master in fictional text games. Generate your next step of action after Action. Action must not be empty. e.g. Action: put down cup.

------------------------------------------------

**Instruction**

You are in a fictional game environment and you need to accomplish goals by performing actions. Each action is a simple phrase. Here are the actions you can do:
- Inventory: check things you are carrying
- Look: check your surroundings

- Examine place/object: check the details of something
- Take object: pickup object
- Put down object: leave a object at your current place.
- Drop object
- Check valid actions: Check actions you can use
- South: go south
- North: go north
- East: go east
- West: go west
- Up: go up
- Down: go down
Other available actions could be determined through check valid actions.

- - - - - - - - - - - - - - - - - - - - - - - - - - - - - - - - - - - - - - - - - - - - - - - - - - - -

**Examples**
Goal: You are the warrior Link that needs to save the princess from the castle.
Observation: You are at the path leading to the castle. The castle is to your north. There is a barrel in front of you.
Action: Examine barrel
Observation: There is a gleaming elvish sword.
Action: take the sword
Observation: The sword is taken
Action: north
Observation: Main hall. The castle is dimly lit, with lava flowing in many places. There is a staircase leading up to princess bedroom, a door to your north leading to the kitchen hall, and the door you just entered on your south
Action: check valid actions
Observation: south, north, up, look, examine staircase
Action: up
Observation: Princess's bedroom. Princess is sleeping in bed.
Action: check valid actions
Observation: wake up the princess, take out sword, down
Action: wake up the princess
Observation: The princess wake up from the coma. Thank you my knight, she says. The task is finished.

Figure 14: Prompt details for Jericho.

---

**Prompt Details for PDDL**

**System Prompt**
You are a master in planning. Generate your next step of action after Action. Action must not be empty. e.g. Action: put down cup.

- - - - - - - - - - - - - - - - - - - - - - - - - - - - - - - - - - - - - - - - - - - - - - - - - - - -

**Instruction**
The robot has four actions: pickup, putdown, stack, and unstack. The domain assumes a world where there are a set of blocks that can be stacked on top of each other, an arm that can hold one block at a time, and a table where blocks can be placed.
The actions defined in this domain include:
pickup `<block>`: allows the arm to pick up a block from the table if it is clear and the arm is empty. After the pickup action, the arm will be holding the block, and the block will no longer be on the table or clear.
putdown `<block>`: allows the arm to put down a block on the table if it is holding a block. After the putdown action, the arm will be empty, and the block will be on the table and clear.
stack `<block>` `<block>`: allows the arm to stack a block on top of another block if the arm is holding the top block and the bottom block is clear. After the stack action, the arm will be empty, the top block will be on top of the bottom block, and the bottom block will no longer be clear.
unstack `<block>` `<block>`: allows the arm to unstack a block from on top of another block if the arm is empty and the top block is clear. After the unstack action, the arm will be holding the top block, the top block will no longer be on top of the bottom block, and the bottom block will be clear.

- - - - - - - - - - - - - - - - - - - - - - - - - - - - - - - - - - - - - - - - - - - - - - - - - - - -

**Examples**
Goal: The goal is to satisfy the following conditions: b1 is on b2., b2 is on b3. Observation: b1 is on

the table. b2 is on the table. B3 is on the table. Robot arm is empty. The b1 is clear. The b2 is clear. The b3 is clear.
Action: pickup b2
Observation: b1 is on the table. B2 is on the table. The b1 is clear. The b3 is clear. You are holding b2.
Action: stack b2 b3
Observation: b1 is on the table. b1 is on b2. B3 is on the table. Robot arm is empty. The b1 is clear. The b2 is clear.
Action: pickup b2.
Observation: The action is not valid and therefore takes no effect. Please remember to satisfy the restriction of actions. You can also check valid actions.
Action: check valid actions.
Observation: valid actions are: pickup b2, unstack b1 b2.
Action: pickup b1
Observation: b2 is on b3. B3 is on the table. Robot arm is empty. The b2 is clear. You are holding b1.
Action: stack b1 b2
Observation: b1 is on b2. b2 is on b3. B3 is on the table. Robot arm is empty. The b1 is clear. The goal is satisfied.

Figure 15: Prompt details for PDDL. The provided instruction/example are changed based on the type of the specific environment instance.

## Prompt Details for WebShop

**System Prompt**
You are a helpful virtual webshop assistant that interacts with the simulated website to solve a task.

- - - - - - - - - - - - - - - - - - - - - - - - - - - - - - - - - - - - - - - - - - - - - - - - - -
**Instruction**
You are now the virtual webshop assistant, navigating a website to locate and purchase items based on given commands. Our interaction will follow this structure:

Your Actions: You will preface each of your actions with "Action: ".
Website's Response: The website will provide feedback starting with "Observation: ".

[click]something: Engage with specific buttons or links.
[search]something: Seek specific data on the website. Use this only if a [Search] button appears in the observation.
Note: If you wish to search and there's no [Search] button, click the [Back to Search] button instead.

Observation Format: The website will showcase its content. Elements within square brackets (like [Buy Now]) indicate clickable buttons or links.

- - - - - - - - - - - - - - - - - - - - - - - - - - - - - - - - - - - - - - - - - - - - - - - - - -
**Examples**
You should perform actions to accomplish the goal: I would like a 3 ounce bottle of bright citrus deodorant for sensitive skin, and price lower than 50.00 dollars
Action: reset[]
Observation: WEB PAGE: {Let us begin shopping
[Search]}

Action: search[3 ounce bright citrus deodorant sensitive skin]
Observation: WEB PAGE: {
[Back to Search]
Page 1 (Total results: 15)
[Next >]
[B078GWRC1J]
Bright Citrus Deodorant by Earth Mama | Natural and Safe for Sensitive Skin, Pregnancy and Breastfeeding, Contains Organic Calendula 3-Ounce
$10.99}

Action: click[B078GWRC1J]
Observation: WEB PAGE: {

```
[Back to Search]
[<Prev]
scent [assorted scents][bright citrus][calming lavender][ginger fresh][simply non-scents]
size [3 ounce (pack of 1)]}

Action: click[bright citrus]
Observation: You have clicked bright citrus.

Action: click[3 ounce (pack of 1)]
Observation: You have clicked 3 ounce (pack of 1).

Action: click[Buy Now]
Observation: You have bought 3 ounce (pack of 1).
```

Figure 16: Prompt details for WebShop.

---

**Prompt Details for WebArena**

**System Prompt**
You are an autonomous intelligent agent tasked with navigating a web browser. You will be given web-based tasks. These tasks will be accomplished through the use of specific actions you can issue.

- - - - - - - - - - - - - - - - - - - - - - - - - - - - - - - - - - - - - - - - - - - - - - - - - - - - -

**Instruction**
Here's the information you'll have:
The user's objective: This is the task you're trying to complete.
The current web page's accessibility tree: This is a simplified representation of the windowed webpage, providing key information.
The current web page's URL: This is the page you're currently navigating.
The open tabs: These are the tabs you have open.

The useful websites and corresponding URL you can navigate:
'reddit': "http://reddit.com"
'online shop': "http://onestopmarket.com"
'e-commerce platform': "http://luma.com/admin"
'gitlab': "http://gitlab.com"
'wikipedia': "http://wikipedia.org"
'map': "http://openstreetmap.org"

The actions you can perform fall into several categories:

Page Operation Actions:
'click [id]': This action clicks on an element with a specific id on the webpage.
'type [id] [content] [press_enter_after = 0 |1]': Use this to type the content into the field with id. By default, the "Enter" key is pressed after typing unless press_enter_after is set to 0.
'hover [id]': Hover over an element with id.
'press [key_comb]': Simulates the pressing of a key combination on the keyboard (e.g., Ctrl+v).
'scroll [directio n= down |up]': Scroll the page up or down.

Tab Management Actions:
'new_tab': Open a new, empty browser tab.
'tab_focus [tab_index]': Switch the browser's focus to a specific tab using its index.
'close_tab': Close the currently active tab.

URL Navigation Actions:
'goto [url]': Navigate to a specific URL.
'go_back': Navigate to the previously viewed page.
'go_forward': Navigate to the next page (if a previous 'go_back' action was performed).

Completion Action:
'stop [answer]': Apply this action when you believe the task is complete. If it is a operation-type task, use 'stop [Done]' when finished. If the objective is to give a text-based answer, provide the answer in

the bracket.

To be successful, it is very important to follow the following rules:
1. You should only issue an action that is valid given the current observation
2. You should only issue one action at a time.
3. Generate the action in the correct format and always put the action inside a pair of @. Such as, @click [1234]@.
4. Complete the task by interacting with the starting page, and avoid using 'goto' actions casually.
5. Reasonable inputs will return accurate observations, so do not repeat the same action when unnecessary.

- - - - - - - - - - - - - - - - - - - - - - - - - - - - - - - - - - - - - - - - - - - - - - - - - -
**Examples**
You should perform actions to accomplish the goal: Add a white desk to my wish list
Observation:
WINDOWED PAGE:{
Tab 0 (current): One Stop Market
[1] RootWebArea 'One Stop Market' focused: True
    [1254] link 'My Wish List'
    [2427] StaticText 'Search'
    [1585] combobox 'Search' autocomplete: both hasPopup: listbox required: False expanded: False
    [2430] link 'Advanced Search'
    [1588] button 'Search' disabled: True
}
URL: http://onestopmarket.com

Action: @type [1585] [white desk] [press_enter_after=1]@
Observation:
WINDOWED PAGE:{
Tab 0 (current): Search results for: 'white desk'
[2635] RootWebArea "Search results for: 'white desk'" focused: True
    [3869] link 'My Wish List'
    [4827] StaticText 'Search'
    [4072] combobox 'Search' autocomplete: both hasPopup: listbox required: False expanded: False
        [5027] StaticText 'white desk'
    [4830] link 'Advanced Search'
    [4075] button 'Search' disabled: True
    [3729] main ''
        [3842] heading "Search results for: 'white desk'"
        [3907] StaticText 'Items 1-12 of 38823'
        [4781] link 'Image'
            [4833] img 'Image'
        [4783] link 'Image'
            [4849] img 'Image'
}
URL: http://onestopmarket.com/catalogsearch/result/?q=white+desk
...

Figure 17: Prompt details for WebArena. The provided example is changed based on the type of the specific environment instance.

---

**Prompt Details for Academia**

**System Prompt**
You can use actions to help people solve problems.

- - - - - - - - - - - - - - - - - - - - - - - - - - - - - - - - - - - - - - - - - - - - - - - - - -
**Instruction**
We detail name, description, input(parameters) and output(returns) of each action as follows:
Name: loadPaperNet()
Description: Load PaperNet. In this net, nodes are papers and edges are citation relationships between papers.

Name: loadAuthorNet()
Description: Load AuthorNet. In this net, nodes are authors and edges are collaboration relationships between authors.

Name: neighbourCheck(graph, node)
Description: List the first-order neighbors connect to the node. In paperNet, neigbours are cited papers of the paper. In authorNet, neigbours are collaborators of the author.
Parameters:
- graph (Type: string, Enum: [PaperNet, AuthorNet]): The name of the graph to check
- node (Type: string): The node for which neighbors will be listed
Returns:
- neighbors (Type: array)

Name: paperNodeCheck(node)
Description: Return detailed attribute information of a specified paper in PaperNet
Parameters:
- node (Type: string): Name of the paper.
Returns:
- authors : The authors of the paper
- year : The puslished year of the paper
- venue : The published venue of the paper
- n_citation : The number of citations of the paper
- keywords : The keywords of the paper
- doc_type : The document type of the paper

Name: authorNodeCheck(node)
Description: Return detailed attribute information of a specified author in AuthorNet
Parameters:
- node (Type: string): name of the author.
Returns:
- name : The name of the author
- org : The organization of the author

Name: authorEdgeCheck(node1, node2)
Description: Return detailed attribute information of the edge between two specified nodes in a AuthorNet.
Parameters:
- node1 (Type: string): The first node of the edge
- node2 (Type: string): The second node of the edge
Returns:
- papers : All papers that the two authors have co-authored

Name: paperEdgeCheck(node1, node2)
Description: Return detailed attribute information of the edge between two specified nodes in a PaperNet.
Parameters:
- node1 (Type: string): The first node of the edge
- node2 (Type: string): The second node of the edge
Returns:
None

Name: check_valid_actions()
Description: Get supported actions for current tool.
Returns:
- actions (Type: array): Supported actions for current tool.

Name: finish(answer)
Description: Return an answer and finish the task
Parameters:
- answer (Type: ['string', 'number', 'array']): The answer to be returned

If you are finished, you will call "finish" action

Please refer to the format of examples below to solve the requested goal. Your response must be in the format of "Action: [your action] with Action Input: [your action input]"

- - - - - - - - - - - - - - - - - - - - - - - - - - - - - - - - - - - - - - - - - - - - - - - - - -

**Examples**
Goal: When was the paper Learning the Principle of Least Action with Reinforcement Learning. published?

Action: loadPaperNet with Action Input: {}
Observation: PaperNet is loaded.
Action: paperNodeCheck with Action Input: {"node":"Learning the Principle of Least Action with Reinforcement Learning."}
Observation: {'year': 2021, 'venue': 'AAAI Spring Symposium - MLPS', 'n_citation': 0, 'keywords': [], 'doc_type': 'Conference'}
Action: finish with Action Input: {"answer": "2021"}
Observation: 2021

Figure 18: Prompt Details for Academia in Tool-Query Environments.

## Prompt Details for Todo

**System Prompt**
You can use actions to help people solve problems.

- - - - - - - - - - - - - - - - - - - - - - - - - - - - - - - - - - - - - - - - - - - - - - - - - -

**Instruction**
We detail name, description, input(parameters) and output(returns) of each action as follows:
Name: get_user_current_date()
Description: Get the user's current date.
Returns:
The current date in 'YYYY-MM-DD' format.

Name: get_user_current_location()
Description: Get the user's current city.
Returns:
The user's current city.

Name: get_projects()
Description: Get all projects in the Todoist account
Returns:
- Array of objects with properties:
- id (Type: string)
- name (Type: string)
- order (Type: integer)
- color (Type: string)
- is_favorite (Type: boolean)

Name: update_project(project_id, is_favorite)
Description: Update a project
Parameters:
- project_id (Type: string)
- is_favorite (Type: string, Enum: [True, False])
Returns:
Information of the updated project

Name: get_tasks(project_id)
Description: Get all tasks for a given project
Parameters:
- project_id (Type: string)
Returns:
- Array of objects with properties:
- id (Type: string)
- project_id (Type: string)

- order (Type: integer)
- content (Type: string): Name of the task.
- is_completed (Type: boolean)
- priority (Type: integer): Task priority from 1 (normal) to 4 (urgent).
- due_date (Type: string): The due date of the task.

Name: get_task_description(task_id)
Description: Get the description of a specific task in the Todoist account.
Parameters:
- task_id (Type: string)
Returns:
- id (Type: string): Unique identifier of the task.
- content (Type: string): Name of the task.
- description (Type: string): Description of the task. Incluing the Place, Tips, etc.

Name: get_task_duration(task_id)
Description: Get the duration of a specific task in the Todoist account.
Parameters:
- task_id (Type: string)
Returns:
- id (Type: string)
- content (Type: string): Name of the task.
- duration (Type: string): Duration of the task in the format of 'amount(unit)'.

Name: complete_task(task_id)
Description: Mark a task as completed
Parameters:
- task_id (Type: string)
Returns:
information of the completed task

Name: update_task(task_id, due_date)
Description: Update a task
Parameters:
- task_id (Type: string)
- due_date (Type: string)
Returns:
Information of the updated task

Name: delete_task(task_id)
Description: Delete a specific task from the Todoist account.
Parameters:
- task_id (Type: string): Unique identifier of the task to delete.
Returns:
Information of the deleted task.

Name: check_valid_actions()
Description: Get supported actions for current tool.
Returns:
Supported actions for current tool.

Name: finish(answer)
Description: Call this action, when find the answer for the current task or complete essential operations.
Parameters:
- answer (Type: ['string', 'number', 'array']): If the task is a question answering task, this is the answer to be returned. If the task is an operation task, the answer in 'done'

If you are finished, you will call "finish" action
Please refer to the format of examples below to solve the requested goal. Your response must be in the format of "Action: [your action] with Action Input: [your action input]"
- - - - - - - - - - - - - - - - - - - - - - - - - - - - - - - - - - - - - - - - - - - - - - - - - - - - -
**Examples**

Goal: Is Prepare for history quiz a task of School project? Please answer yes or no.

Action: get_projects with Action Input: {}
Observation: [{'id': '12345', 'order': 0, 'color': 'charcoal', 'name': 'School', 'is_favorite': false}]
Action: get_tasks with Action Input: {"project_id": "12345"}
Observation: [{'id': '123451', 'order': 0, 'content': 'Prepare for history quiz', 'is_completed': false, 'priority': 1, 'due_date': '2030-10-10'}, {'id': '123452', 'order': 1, 'content': 'Prepare for math quiz', 'is_completed': false, 'priority': 1, 'due_date': '2030-11-10'}]
Action: finish with Action Input: {"answer": "yes"}
Observation: yes

Figure 19: Prompt Details for Todo in Tool-Operation Environments.

