# OpenReview forum: "AgentBoard: An Analytical Evaluation Board of Multi-turn LLM Agents"
_NeurIPS.cc/2024/Datasets_and_Benchmarks_Track — NeurIPS 2024 Track Datasets and Benchmarks Oral_

### Official Review · Reviewer_sNFC · 2024-07-24
**Initial Review of AgentBoard Benchmark Paper**

**Rating:** 7
**Confidence:** 4
**Correctness:** The experiment setup in the paper is …
**Clarity:** The paper is well written and easy to…

**Review:**

The benchmark provides a valuable tool for evaluating AI agents' planning abilities to call external functions and accomplish complex tasks.

I tested several LLM models in four of the environments in this benchmark and found the codebase easy to use and of high quality. I believe the paper is above the acceptance threshold, though it has clear pros and cons.

Pros:
1. The codebase is easy to use and well-maintained.
2. The benchmark includes a visualization tool for advanced analysis, such as progress gain relative to trajectory length up to 30 steps.
3. The tasks are diverse, ranging from embodied AI tasks to e-commerce tasks, and are well-integrated into the unified framework.

Cons:
1. The number of test cases is too small in some scenarios (e.g., only 20 for JC and 40 for TO), leading to potentially drastic variations in evaluation results, which may be unconvincing or even misleading.
2. The action space is too limited in all environments except Game-Jericho. However, since JC only has 20 test cases, the benchmark cannot effectively test models' abilities in complex action spaces.
3. The subgoals in each test case are annotated by humans, and the paper does not reveal clear rules for subgoal design, making it challenging to include new test cases.

**Strengths:**

As discussed above, the paper has several clear strengths as a benchmark, including:

1. A high-quality codebase that is well-maintained.
2. Effective visualization tools for advanced analysis.
3. Diverse task scenarios integrated into a unified framework.

Additionally, evaluating the multi-turn functional call abilities of LLM-based agents is an important task and of significant interest to the AI agent community.

**Additional Feedback:**

It is unclear to me how each annotated sub-goal in the sequence is ensured to be necessary. Is it possible that the LLM could achieve the final goal without passing through some of the sub-goals, potentially leading to a false negative evaluation?

**Documentation:**

The paper provides links to the data collection resources. The results are reproducible.

**Limitations:**

The paper already properly addressed several limitations.

Additionally, I think another limitation is the limited numbers of test cases in some environments. Evaluating models on scenarios with too few test cases may lead to misleading results and negatively impact model evaluation.

**Opportunities For Improvement:**

1. The number of test cases in some scenarios, especially for JC, TO, and TQ, is too small. Including more test cases would significantly enhance the value and reliability of the evaluation benchmark.
2. The paper could better address the rules and intuitions behind the design. For example, more detail on how the sequence of tasks is determined and how subgoals are designed would provide greater clarity and allow for more consistent and replicable test case creation.

**Relation To Prior Work:**

A table is provided to represent the differences of this paper to previous works.

**Summary And Contributions:**

AgentBoard is a benchmark designed to test LLM-based agents on their multi-turn functional call abilities. It integrates nine environments within four diverse task categories into one unified testing framework. The agents' abilities are measured by the success rate (whether the final goal is accomplished) and the progress rate (based on human-annotated subgoals). Additionally, it provides a visualization tool for more detailed analysis.

The paper is valuable for evaluating AI agents' tool-use abilities and long-term planning capabilities. This is an emerging and important research area in LLM-based agents.

---

> ### Author Rebuttal · Authors · 2024-08-17
>
> Thanks for your valuable suggestions! We address the concerns below.
>
> > **Q1: The number of test cases is too small in some scenarios, ​​leading to potentially drastic variations in evaluation results**
>
> A1: This is a valid point. The number of test cases for some subtasks is limited partially due to expensive annotation on these environments, and we agree with the reviewer that the score variance may be high on the single task with a very small number of test cases. Despite this limitation, we highlight that on the higher-level category separation, each category (e.g. Game or Tool) has over 100 test cases, and thus the average scores within each category, or the overall average scores for the entire AgentBoard, can be used to reflect agent performance much more stably. In the next revision, we plan to report the average scores within each category (which can be easily computed with the current per-task scores and the number of test cases of each task) as well for a more stable comparison to be potentially referenced by future researchers. In the meanwhile, we do plan to add more test examples to the tasks with a small number of test cases in future maintenance of AgentBoard.
>
>
> > **Q2: The action space is too limited**
>
> A2: Our type of actions/APIs is relatively limited, with only 8-15 action types. However, generating the correct and logical action remains challenging. The agent must determine not only the type of action but also the appropriate argument from the parameter set. For instance, the action "put {obj a} on {obj b}" can result in many combinations based on the available objects. In tasks like ScienceWorld, there are about 100 valid actions to choose from for each step. Although the action space is small, it is complex. The best-performing GPT-4 model achieves only 86% grounding accuracy, while the top open-source model, Llama3-70B, has just 66%. This shows that it is already challenging for models to use these APIs correctly.
>
>
>
>
>
> > **Q3: The paper could better address the rules and intuitions behind the design.**
>
> A3: Thanks for the advice! During the annotation stage, a human player uses an interactive annotation panel to first engage with the task until finishing. The panel records all states and actions and the player should select essential states that must be passed through and record them as subgoals. A subsequent verification process ensures quality of these annotations as introduced in Appendix I. We will provide more details and clarify these designs further in the next revision for more consistent and replicable test case creation.

---

### Official Review · Reviewer_HfY5 · 2024-07-25
**Reviews**

**Rating:** 7
**Confidence:** 3
**Clarity:** Yes, the logic and structure of this …

**Review:**

This article has clear logic and a well-defined structure, emerging as a high-quality benchmark after reviewing numerous papers in the relevant field. The most important original contribution is the design of the progress rate metric, with the analysis of LLMs' capabilities primarily reflected in the comparison of this metric. While most other works use success rate as the sole evaluation criterion, this article indeed fills this gap. For pros, please refer to the strengths listed below. For cons, refer to the limitations.

**Strengths:**

1. This benchmark includes nine diverse environments across four scenarios with multi-round interactions. This design ensures the diversity of tasks and can better evaluate the planning capabilities of LLMs.
2. They propose a fine-grained progress rate metric tracking the intermediate progress of different agents. This metric can better reflect the differences in capabilities between LLMs. Moreover, calculating this metric requires manual annotation of many appropriate subgoals. Although this consumes considerable resources upfront, it indeed provides a better standard for evaluating the capabilities of LLMs.
3. AgentBoard provides a comprehensive evaluation of various aspects of different LLMs' capabilities, with thorough experiments conducted.

**Additional Feedback:**

In Table 5, it is reasonable that the decrease in progress for hard tasks, compared to easy tasks, is less significant than the decrease in success. For example, the experimental results for GPT-4 conform to this observation. However, in the experimental results for other models, such as Lemur-70b, many opposite results appear. Could you explain the reason for this phenomenon?

**Correctness:**

The evaluation methods and experiment design are appropriate and performed correctly. However, there are also some issues present. For details, please refer to the limitations.

**Documentation:**

Yes, this article provides sufficient details to support reproducibility.

**Ethics:**

No.

**Limitations:**

1. The most important limitation, as the authors also mentioned in the paper, is the need for manual annotation of subgoals. Other works have to rely solely on success rate as the final evaluation criterion due to the lack of intermediate subgoals. While this paper has invested considerable resources to address this deficiency and has indeed brought benefits, it is detrimental to the extensibility of the benchmark. For instance, when attempting to use other tasks to evaluate the performance of LLMs, the progress rate metric cannot be computed.
2. Does imposing restrictions on tools and locations to achieve a unique path limit the capabilities of LLMs? For instance, without considering subgoals, would removing the restrictions on locations and tools potentially increase the success rate of LLMs? If the answers to these two questions are yes, then the experimental results are limited.

For other limitations, please refer to Opportunities For Improvement.

**Opportunities For Improvement:**

1. In the future work, it may be beneficial to include additional datasets, such as MC-TextWorld, which is also used in the paper "Describe, Explain, Plan and Select: Interactive Planning with Large Language Models Enables Open-World Multi-Task Agents". After understanding the purpose of this benchmark, I believe that incorporating the open-world game MC from the aforementioned study would be well-suited for evaluating the capabilities of LLMs in more complex environments.
2. Whether training a specialized model to generate subgoals instead of relying on manual annotation an available approach? After all, this part consumes significant human resources, and the final evaluation results heavily depend on the human annotations.

**Relation To Prior Work:**

Yes, the differences and advantages of this article compared to previous works are clearly described.

**Summary And Contributions:**

This paper introduces AgentBoard, a pioneering comprehensive benchmark and accompanying open-source evaluation framework tailored to an analytical evaluation of LLM agents. AgentBoard offers a fine-grained progress rate metric as well as a comprehensive evaluation toolkit. It improves the interpretability of the performance of LLMs greatly. This benchmark includes a variety of tasks and carefully designed metrics that help comprehensively evaluate the capabilities of LLMs.

---

> ### Author Rebuttal · Authors · 2024-08-17
>
> Thanks for the valuable suggestions! We address the questions below.
>
> > **Q1:  In the future work, it may be beneficial to include additional datasets, such as MC-TextWorld**
>
> A1: Thanks for the great suggestion! We agree that incorporating the open-world game MC will be well-suited for our benchmark to evaluate LLMs in more complex environments. We’ll definitely consider adding it in future maintenance
>
> > **Q2: Whether training a specialized model to generate subgoals instead of relying on manual annotation is an available approach?**
>
> A2: This is a good idea and it could help extend the benchmark more easily. We think that generating subgoals automatically may be feasible and some previous studies [1][2] have used LLMs like GPT-4 to label subgoals, which can effectively guide LLMs as value functions. We resorted to human annotations to ensure correctness of these subgoals to be used during benchmarking, and we will explore automatic generation of these subgoals as future work when we further expand AgentBoard.
>
>
> [1] Zhang, Shenao, et al. "How Can LLM Guide RL? A Value-Based Approach." arXiv preprint arXiv:2402.16181 (2024).
>
> [2] Liu, Zhihan, et al. "Reason for future, act for now: A principled framework for autonomous llm agents with provable sample efficiency." ICML 2023.
>
>
> > **Q3: The most important limitation, as the authors also mentioned in the paper, is the need for manual annotation of subgoals.**
>
> A3: We use human annotations to ensure the correctness of subgoals during benchmarking, prioritizing reliability over scalability. We agree that human annotation is time-consuming and limits extensibility of the benchmark, thus we plan to explore automatic subgoal generation as described in A2 above when we further expand AgentBoard in the future.
>
> > **Q4: Does imposing restrictions on tools and locations to achieve a unique path limit the capabilities of LLMs? For instance, without considering subgoals, would removing the restrictions on locations and tools potentially increase the success rate of LLMs?**
>
> A4: Imposing restrictions on tools and locations will not lower the performance of LLMs, as the main requirement is for the LLM to comprehend the instructions. We tried to ensure that the designated tool is not more difficult to use than another tool. For example, using a pot on a stove to heat is not more challenging and does not require more steps than using a microwave.  In contrast, we observe slight improvement of success rate after giving the model restriction instructions, as the restriction can act as a hint to the model; for instance, after giving the model restriction instructions *“You should get apple juice and boil it in the kitchen. The objects you can use are a metal pot, a thermometer, a fridge, cups, and a stove in the kitchen. ”*, it can learn from instructions to use the stove to heat an object.

---

> > ### Comment · Reviewer_HfY5 · 2024-08-27
> > **Thank you for the responses**
> >
> > Thank you for the responses.

---

> > > ### Author Response · Authors · 2024-08-27
> > > **Thank you for your reply**
> > >
> > > Thank you for your timely feedback and helpful suggestions! We hope we have addressed your concerns and are here to answer any further questions you may have.

---

### Official Review · Reviewer_h7Ld · 2024-07-29
**Analytical Evaluation Board of Multi-Turn LLM Agents**

**Rating:** 7
**Confidence:** 3
**Clarity:** The paper is clear, well written and …

**Review:**

Overall, this paper is clear, well-motivated, and provides a new benchmark with progress rate metrics that allow more fine-grained analytical evaluation of AI agent capabilities. Most benchmarks use success rate as their metric, and many attempt to break down the tasks into small subtasks to allow more granular evaluation in agents' various cognitive capabilities. However, there are not many attempts to define a more analytically rigorous metric for multi-round, partially observable environment. This paper paves the way for more research in quantitative metrics that allow more granular, analytical evaluation of agent performances that will contribute to the academic community and industry as a whole.

However, there are limitations to the paper, specifically that progress rate metrics are largely task dependent and currently requires manual work to break into subgoals. Although it is possible to use LLMs to generate the subgoals (commonly done to evaluate agent planning capabilities), the current approach is potentially more reliable, accurate but lacks scalability. It would be interesting to see if there are alternatives that allow the generic, accurate creation of task-specific sub-goals for evaluation of progress. Further, all tasks are based on text interaction only, which is rather limiting given the general expectation of multi-modal AI agents.

**Strengths:**

The paper is clearly written, well motivated, with detailed documentation, visualization and leaderboard available. It proposes a novel progress rate metric that allows more granular, analytical evaluation of agent capabilities that will greatly contribute to the broader research community. It also provides a fairly diverse set of tasks, in embodied world, game, web and tool use, with multi-round, partially observable environments which are closer to real-life tasks.

**Additional Feedback:**

Typo: in the supplementary material - data card page 1, "Multi-round Intercation" should be "Multi-round Interaction".

**Correctness:**

To the best of my knowledge, there are no correctness issue regarding benchmark evaluation methods and experiment design. I have also added other comments in the sections above.

**Documentation:**

The authors provided sufficient details, including code and GitHub repository to support reproducibility. They also provided a leaderboard with clear visualization of model performance.

**Ethics:**

There are no further ethical concerns regarding this paper to the best of my knowledge.

**Limitations:**

The authors addressed the main limitation of the paper, which is the lack of scalability of generating task-specific progress rate metric. They have adequately addressed the limitations of their research.

**Opportunities For Improvement:**

The paper could benefit from discussions with regards to the following points:
1. The paper is not very clear about the definition of grounding, for which there is no universal definition / benchmark yet. Even though the general task success rate is relatively low, there is almost saturation already in the grounding accuracy metric. Does that mean the agents are already great at planning / segmentation?
2. All tasks are fully text based, which is rather limiting compared with the general expectation for multi-modal AI agents.
3. In the paper, the authors discussed the results and reached the conclusion that code skills help with agent tasks - which makes sense for tasks involving coding and functional calls, but there is less evidence on improvement of general planning capabilities due to slightly better results in game environment with 2 finetuned code models.
4. As the paper lists quite a few models running through the benchmark, it might be clearer to sort results based on model size / parameter numbers for comparisons.
5. Given that sub-goals are defined by humans, it is sometimes difficult to distinguish between subgoals and simply steps to accomplish the tasks. Are there any particular guidelines on how sub-goals are created?

**Relation To Prior Work:**

To the best of my knowledge, this work discusses how it differs from previous contributions.

**Summary And Contributions:**

This paper proposes AgentBoard, a new benchmark with human-labelled progress rate metrics that allow more fine-grained analytical evaluation of AI agent capabilities. It also incorporates diverse tasks, multi-round interactions, partially observable environments, and clear documentations and visualizations for evaluating generalist LLM agents.

---

> ### Author Rebuttal · Authors · 2024-08-17
>
> Thanks for the positive review! We address the concerns below.
> > **Q1: The paper is not very clear about the definition of grounding, for which there is no universal definition / benchmark yet**
>
> A1:  Definition of grounding indeed varies in different works. In this work, we adopt the definition from [1][2], which describes grounding as the process of mapping high-level plans to **executable** actions, no matter whether the plan is correct or wrong. This requires the LLM's ability to format outputs according to instructions, which is crucial for both tool usage and robotic tasks. While grounding accuracy is improving as LLMs become adept at generating API calls and formatting outputs, success in grounding does not necessarily indicate that agents excel at planning, as executable actions do not imply the high-level plans are correct. Even if all actions are executable, a problem may remain unsolved due to logical or decision-making errors.
>
> [1] Gu, Yu, Xiang Deng, and Yu Su. "Don’t Generate, Discriminate: A Proposal for Grounding Language Models to Real-World Environments." ACL. 2023.
>
> [2] Zhong, Victor, et al. "SILG: the multi-environment symbolic interactive language grounding benchmark." NeurIPS 2021.
>
> > **Q2: All tasks are fully text based, which is rather limiting compared with the general expectation for multi-modal AI agents.**
>
> A2: Thank you for your feedback! We agree that multimodal agent evaluations are important for LLM agents. Our submission focuses on evaluating agentic ability of text LLMs, and we will explore multimodal versions in future works.
>
> > **Q3: code skills help with agent tasks - which makes sense for tasks involving coding and functional calls, but there is less evidence on improvement of general planning capabilities due to slightly better results in game environment**
>
> A3: We conclude that code skills help with agent tasks due to the substantial average improvement when comparing Lemur-70b & CodeLlama-34b to Llama2-70b, and comparing CodeLlama-13b to Llama2-13b. The code models outperform their chat counterparts across most tasks. We agree with the reviewer that substantial improvements are not uniformly observed across all tasks, thus we plan to revise this claim to make it more accurate in the next revision of the paper.
>
>  > **Q4: it might be clearer to sort results based on model size / parameter numbers for comparisons**
>
> A4: Thanks for the suggestion! We will provide clearer comparison of different model sizes.
>
> > **Q5:   it is sometimes difficult to distinguish between subgoals and simply steps to accomplish the tasks. Are there any particular guidelines on how sub-goals are created?**
>
> A5: The main difference between subgoals and steps is that subgoals serve as necessary checkpoints for the task. We ensure that the agent must accomplish these subgoals to complete the task, although it may take various steps, including detours, to reach them.
>
> We annotate subgoals using an interactive panel that requires the human player to play through the task first. The panel records the history of actions and states, as well as tell whether the player has finished the task. The player then selects the essential states from the history that must be passed through and records them as subgoals. Then a rigorous human verification process is performed described in Appendix I to ensure quality of these subgoals.

---

> > ### Comment · Reviewer_h7Ld · 2024-08-28
> >
> > Thanks for the detailed response and I maintain my current rating to accept this paper.

---

> > > ### Author Response · Authors · 2024-08-28
> > > **Thank you for your response**
> > >
> > > Thank you for your response and again for your valuable suggestions !

---

### Official Review · Reviewer_n2d6 · 2024-07-30
**Review comments for AgentBoard**

**Rating:** 8
**Confidence:** 5
**Correctness:** Partially correct
**Clarity:** Yes

**Review:**

**Strengths**:

1. **Ease of Use**: Having personally run the benchmark, I found that AgentBoard is much easier to set up and use for conducting evaluations.

2. **Multi-round Interactions & Verifications**: It features multi-round trajectories, which are authenticated through human verification, enhancing the reliability of the evaluations.

3. **Wide Coverage of environments**: AgentBoard encompasses nine environments with diverse tasks, including web interactions, tool usage, embodied AI, and gaming.

4. **Additional Evaluation Metric**: The inclusion of a progress rate metric allows for more detailed performance tracking of LLM agents in intermediate turns.

5. **Insightful Analysis**: The analysis of current commercial models and open-source LLM agents offers valuable insights for the development of future agent models and environments.

**Weaknesses**:

1. **Limited Test Case Diversity to Reflect Agent Performance**: Many environments have only a few test cases, which is not diverse enough to accurately reflect agent performance. For example, Jericho has only 20 test cases, tool-operation has 40, and tool-query includes three distinct environments, but each environment has only 20 test cases.

2. **Lack of Evaluation Detail**: The paper lacks detailed explanations of how the metrics are calculated for each environment, making it difficult to understand the evaluation process fully.

3. **Limited LLM Agent Baselines**: Among the selected open-source LLMs, only Lemur-70b is specifically designed for agents.

For example, here are two findings regarding to two of the environments

**For toolquery**:

The subgoal evaluation is `frequently fixating upon the exact matching` of the api output with the labeled expected output, it is not taking into account the variability of the API outputs.

Secondly, the progress rate calculation is `only calculating the progress` if the subgoals are achieved 1-after-another without any miscellaneous API callings in between. Hence even if the final output is achieved with some extra API calls, the success rate for the query is measured to be zero.

The `subgoal evaluation seems to be too strict`, restricting the models capability of having any prior information of the query.

The `prompting technique` also does not provide specific details on strictly deciphering from the agent interactions only.

**For webshop**:

The webshop evaluation is **not clear**, their is not much detail on the additional_info category provided and the progress rate calculated.

**Strengths:**

See Strengths.

**Additional Feedback:**

1. Besides the weaknesses mentioned, could you also provide more comments on the two example environments?

2. Among the selected open-source LLMs, only Lemur-70b is specifically designed for agents, while other models like Mistral-7b are general-purpose LLMs. It would be very beneficial to compare with recent LLM agent models, which were released well before the submission deadline:

AgentLM: https://huggingface.co/THUDM/agentlm-70b

xLAM-v0.1-r : https://huggingface.co/Salesforce/xLAM-v0.1-r

**Documentation:**

Yes

**Limitations:**

Yes

**Opportunities For Improvement:**

See weakness

**Relation To Prior Work:**

Yes

**Summary And Contributions:**

Building a unified framework to evaluate large language agent models across diverse tasks, particularly in partially observable environments, presents significant challenges. Additionally, designing multi-round interactions for realistic scenarios is more critical than relying on many existing single-round benchmarks. Currently,  many evaluation benchmarks primarily focus on the final success rate metric, which can be limiting when agent models exhibit extremely low success rates in handling complex environments. This approach often provides limited insights into intermediate performance.

To address these issues, this work introduces AgentBoard, a benchmark that includes nine different environments and tasks such as web interactions, tool usage, embodied AI, and gaming. AgentBoard carefully designs subgoals and incorporates partially observable characteristics. It also introduces a progress rate metric to track the detailed interaction performance of agents. AgentBoard evaluates several commercial and open-source LLM agents, revealing interesting findings.

---

> ### Author Rebuttal · Authors · 2024-08-17
>
> We thank the reviewer for the positive comments! We address the concerns below:
>
> >  **Q1: Limited Test Case Diversity to Reflect Agent Performance**
>
> A1: This is a valid point. The number of test cases for some subtasks is limited, partially due to the expensive annotation of these environments. However, we highlight that in the higher-level category separation, each category (e.g., Game or Tool) has over 100 test cases that are much more diverse. Thus, the average scores within each category can be used to reflect agent performance in a more diverse setting. In the next revision, we plan to report the average scores within each category (which can be easily computed with the current per-task scores and the number of test cases for each task) as well for a more stable comparison, which can potentially be referenced by future researchers. Meanwhile, we plan to add more test examples to the tasks with a small number of test cases in future maintenance of AgentBoard.
>
> > **Q2: Lack of Evaluation detail**
>
> A2:  Thanks for the advice to add more evaluation details!  In general, we compute the progress rate metrics using Equation 2 of the submission, where the specific $r_t^{\text {subgoal}}$ and $r_t^{\text {match}}$ for each environment are detailed in Appendix K. The final, binary success rate is 1 only when progress rate is equal to 1. We briefly summarize the metric computations in the table below, and we will present it more clearly in the next revision of the paper.
>
> | Tasks | Progress Rate                                                                                                                                                                                                                                                                                                                                                                                                                             | Success Rate        |
> |-|-|-|
> | ALF   | We provide subgoal annotations. The progress rate is calculated as $r_t^{\text {subgoal }}$, which is the proportion of  subgoals accomplished.                                                                                                                                                                                                                                                                                           | (Progress rate ==1) |
> | SW    | Similar to ALF.                                                                                                                                                                                                                                                                                                                                                                    | (Progress rate ==1) |
> | BA    | Similar to ALF.                                                                                                                                                                                                                                                                                                                                                                    | (Progress rate ==1) |
> | JC    | Similar to ALF.                                                                                                                                                                                                                                                                                                                                                                      | (Progress rate ==1) |
> | PL    | Progress rate is calculated as $r_t^{\text {match }}$, which is the proportion of goal properties satisfied, the goal properties are explained in Appendix K.6.                                                                                                                                                                                                                                                                           | (Progress rate ==1) |
> | WS    | The progress rate is the average score across three distinct but essential stages of the shopping process, with each state's score based on $r_t^{\text {match }}$. At each stage, the score reflects how closely the selected product matches the target product's attributes and options. The overall progress rate reflects the agent's progress toward successfully purchasing the target product. More details are in Appendix K.7.  | (Progress rate ==1) |
> | WA    | Progress rate is calculated as $r_t^{\text {match }}$, which follows equation 5 in Appendix K.8. The matching function follows the implementation of the original Webarena.                                                                                                                                                                                                                                                               | (Progress rate ==1) |
> | TQ    | Similar to ALF.                                                                                                                                                                                                                                                                                                                                                                      | (Progress rate ==1) |
> | TO    | Progress rate for Sheet task is calculated as $r_t^{\text {match }}$, which is the proportion of cells in the current spreadsheet that align with the golden spreadsheet.  For the Todo task, the progress rate is calculated as $r_t^{\text {subgoal }}$ similar to that in ALF.                                                                                                                                                         | (Progress rate ==1) |

---

> > ### Author Rebuttal · Authors · 2024-08-17
> >
> > >**Q3: Limited LLM Agent Baselines**
> >
> > A3: Thanks for suggesting the AgentLM and xLAM baselines! We run additional experiments to obtain their results as shown below. Both models are strong open-source models. After adding the results, Llama3-70b is still the best open-source model, while xLAM-v0.1-r and AgentLM rank the 2nd and 4th respectively among open-source models. xLAM-v0.1-r’s performance is on par with Llama3-70b and surpasses its base Llama2-70b by 16.4 points in terms of progress rate and 15.9 points in terms of success rate. AgentLM, based on Llama2-70b, surpasses Llama2-70b by 9.5 points in terms of progress rate and 10.2 points in terms of success rate.
> >
> >
> >
> > | Model           | ALF      | SW       | BA       | JC       | PL       | WS       | WA       | TQ       | TO       | Avg      |
> > |-|-|-|-|-|-|-|-|-|-|-|
> > | GPT-4           | 65.5/43.3| 78.8/52.2| 70.7/56.2| 52.4/35.0| 81.2/61.7| 76.5/39.0| 39.4/15.1| 85.1/68.3| 80.8/60.0| 70.0/47.9|
> > | Llama3-70b      | 29.6/12.7| 30.4/7.8 | 41.1/27.7| 16.0/5.0 | 32.2/20.0| 74.6/29.9| 35.6/12.6| 52.6/36.7| 65.2/30.0| 41.9/20.2|
> > | xLAM-v0.1-r 70b | 53.4/42.5| 15.4/1.1 | 37.7/28.6| 16.2/5.0 | 38.4/16.7| 73.6/32.7| 34.5/11.3| 66.5/38.3| 26.5/7.5 | 40.2/20.4|
> > | DeepSeek-67b    | 34.5/20.9| 36.1/10.0| 31.7/22.3| 13.7/0.0 | 22.0/6.7 | 72.7/31.9| 23.9/5.7 | 71.4/40.0| 40.5/17.5| 38.5/17.2|
> > | AgentLM - 70b   | 58.4/50.7| 13.0/1.1 | 38.0/27.7| 8.8/0    | 13.0/3.3 | 72.9/31.1| 13.0/5.3 | 50.5/13.3| 31.8/0.0 | 33.3/14.7|
> > | Llama2-70b      | 13.2/3.0 | 2.6/0.0  | 30.0/19.6| 7.8/0.0  | 8.1/1.7  | 53.6/13.1| 11.6/3.3 | 48.3/0.0 | 38.6/0.0 | 23.8/4.5 |
> >
> >
> > > **Q4: could you also provide more comments on the two example environments? (for the toolquery example)**
> >
> > A4:
> > > 1) **“frequently fixating upon the exact matching”**
> >
> > When constructing APIs for tool-query, we ensure that the functionality and outcome of one action cannot be replicated by combining other actions. This design choice results in a deterministic set of essential actions, which are processed as subgoals. Each subgoal corresponds to the result of an API call.
> >
> > For APIs other than `finish`, the outputs are derived from querying the underlying database. Given the expected input arguments, the output format and content are fixed, allowing for exact matching without issues. However, for the `finish` API, the output is generated by the LLM through summarizing previous interactions. This can potentially affect exact matching. To address this, we incorporated constraints into the instructions. For example, in a test case like: *“On the 19th or the 25th, I want to choose a day to go hiking. Which day has less rainfall? Please provide the answer in the YYYY-MM-DD format”*, we specify the “YYYY-MM-DD” format to reduce variability.
> >
> > In practice, we have not observed any exact matching errors in calculating progress rates.
> >
> > >2) **“even if the final output is achieved with some extra API calls, the success rate for the query is measured to be zero”**
> >
> > We clarify that all subgoals represent necessary intermediate states that must be reached in a fixed order. For example, the agent must first check the movie ID and then the cast to find the director. The task is considered complete once both the movie ID and cast information are obtained, and an answer is correctly given. However, the agent is allowed to take detours or make additional API calls between subgoals. As a result, the success rate will be accurately measured, even if the final output is accompanied by extra API calls.
> >
> > >3) **“The subgoal evaluation seems to be too strict, restricting the models capability of having any prior information of the query”**
> >
> > We aimed for a strict subgoal evaluation because we want the model to complete tasks based solely on agent interactions, rather than relying on prior knowledge. While the model might choose to use its prior knowledge to answer questions, we address how we restrict this in the next reply.
> >
> > > 4) **“The prompting technique also does not provide specific details on strictly deciphering from the agent interactions only. ”**
> >
> > You are right that we did not strictly restrict the models to generate responses *only* based on interaction results, yet we do instruct them to use the provided tools to find answers. We also include relevant in-context examples to guide them. Below is the prompt we use:
> >
> > ```
> > {
> > "system_msg": "You should use actions to help people solve problems.\n"
> >   "instruction":
> > "We detail name, description, input(parameters) and output(returns) of each action as follows:
> > Name: get_search_movie(movie_name)
> > Description: Search for a movie by name and return basic details
> > Parameters:
> > 	- movie_name (Type: string): The name of the movie to search for.
> > Returns:
> > 	- id : The ID of the found movie.
> > 	- overview : The overview description of the movie.
> > 	- title : The title of the movie.
> > …
> > Name: finish(answer)
> > Description: Return an answer and finish the task
> > Parameters:
> > 	- answer (Type: ['string', 'number', 'array']): The answer to be returned
> > If you are finished, you will call \"finish\" action
> > Please refer to the format of examples below to solve the requested goal. Your response must be in the format of \"Action: [your action] with Action Input: [your action input]\""
> > }
> >
> > [example here] …
> >
> > ```
> > Despite the current lenient prompt restrictions, empirically we find that models primarily access information via tools to deliver accurate answers. Besides, our test examples pertain to relatively recent knowledge obtained through tool queries, and in most cases the benchmarked models do not have internal knowledge to address the queries. When curating these examples, we ensure they cannot be directly answered using GPT-4 alone. For instance, in the movie task, we ask about the 2023 Oscar winner *"Everything Everywhere All at Once"*, which falls beyond the knowledge cut-off date for most popular models.

---

> > > ### Author Rebuttal · Authors · 2024-08-17
> > >
> > > > **Q5:  could you also provide more comments on the two example environments? (for the Webshop example environment)**
> > >
> > > A5: The **addition_info** field in the data points provides extra details that are not covered by the main fields. For example, in PDDL tasks, it includes content like {"subtask" : "gripper"}, indicating the specific subtask category. However, in Webshop, this field is not included because this scenario is a relatively straightforward and simple shopping scenario where the agent only needs to select and purchase products based on the requirements mentioned in the instruction. Other main fields including “id”, “goal”, “subgoals” and “difficulty” have already covered all the necessary information.
> > >
> > > The progress rate is calculated as the average score across three stages: search, product selection, and order confirmation. The score for each stage is calculated as follows: on the search result page, the score is the highest among the products returned by the search; on the product description page, the highest score among different options for the product is considered; and on the order confirmation page, the score of the final selected product is used. For a detailed explanation, please refer to Appendix section K.7.

---

> > > > ### Comment · Reviewer_n2d6 · 2024-08-28
> > > > **Reviewer comment for the rebuttal**
> > > >
> > > > Dear Authors,
> > > >
> > > > Thank you for the detailed explanations and additional added results. Please incorporate all of these into the new version. I believe this work is valuable for the agent community, so I have increased my score to "clear accept" to encourage you to continue improving and refining the GitHub code and functions.

---

> > > > > ### Author Response · Authors · 2024-08-28
> > > > > **Thank you for the response**
> > > > >
> > > > > Thank you for your valuable feedback to help us refine the quality of our paper. We will further polish the paper in the final revision. Thank you!

---

### Decision · Program_Chairs · 2024-09-26

**Decision:**

Accept (Oral)

**Comment:**

The reviews are all positive for this paper and it has been well received and noted as easy to engage with making it a good asset for the research community.